# M⁴LE: A Multi-Ability Multi-Range Multi-Task Multi-Domain Long-Context Evaluation Benchmark for Large Language Models

## Abstract

Managing long sequences has become an important and necessary feature for large language models (LLMs). However, it is still an open question of how to comprehensively and systematically evaluate the long-sequence capability of LLMs. One of the reasons is that conventional and widely-used benchmarks mainly consist of short sequences. In this paper, we propose M⁴LE, a **M**ulti-ability, **M**ulti-range, **M**ulti-task, **M**ulti-domain benchmark for **L**ong-context **E**valuation. M⁴LE is based on a diverse NLP task pool comprising 36 NLP datasets, 11 task types and 12 domains. To alleviate the scarcity of tasks with naturally long sequences and incorporate multiple-ability assessment, we propose an automatic approach (but with negligible human annotations) to convert short-sequence tasks into a unified long-sequence scenario where LLMs have to identify single or multiple relevant spans in long contexts based on explicit or semantic hints. Specifically, the scenario includes five different types of abilities: (1) explicit single-span; (2) semantic single-span; (3) explicit multiple-span; (4) semantic multiple-span; and (5) global context understanding. The resulting samples in M⁴LE are evenly distributed from 1k to 8k input length.[1] We conducted a systematic evaluation on 11 well-established LLMs, especially those optimized for long-sequence inputs. Our results reveal that: 1) Current LLMs struggle to understand long context, particularly when tasks require multiple-span attention. 2) Semantic retrieval task is more difficult for competent LLMs. 3) Models fine-tuned on longer text with position interpolation have comparable performance to those using Neural Tangent Kernel (NTK) aware scaling methods without fine-tuning. We make our benchmark publicly available to encourage future research in this challenging area.

## 1 Introduction

Large language models (LLMs) are gaining traction in addressing diverse NLP challenges. LLMs, mostly transformer-based models (Vaswani et al., 2017), are trained on a large amount of data with numerous parameters (Ouyang et al., 2022; Touvron et al., 2023b). These models have demonstrated impressive capabilities across a wide range of tasks (Brown et al., 2020; Schick et al., 2023; Shen et al., 2023; Bang et al., 2023). As LLMs continue to evolve, their ability to handle long-sequence tasks, such as extracting specific information from or summarizing lengthy documents, has become an important and competitive feature (Du et al., 2022; Chiang et al., 2023; Li et al., 2023). Therefore, a comprehensive, fair, and objective benchmark to evaluate the long-sequence capabilities of models is necessary for the progress of LLMs.

Despite numerous efforts to develop benchmarks for assessing the knowledge or reasoning ability of LLMs (Hendrycks et al., 2021; Huang et al., 2023; Suzgun et al., 2022), comprehensive evaluation of their long-context understanding ability has received limited attention. Recent concurrent works, such as L-Eval (An et al., 2023) and LongBench (Bai et al., 2023), primarily rely on existing long-sequence NLP datasets which usually limit the task diversity and flexibility in conducting

---

[1]The released benchmark would contain samples up to 32K words. Even longer samples and other types of tasks can be constructed using our method.

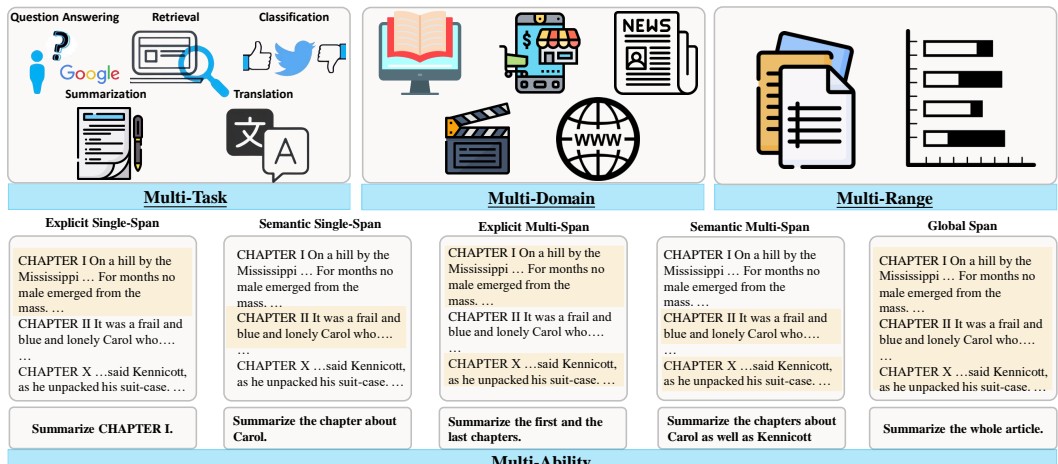

Figure 1: The illustration of M⁴LE. M⁴LE covers multiple task types, domains and length ranges, and introduces five long-context understanding abilities, each of which is exemplified with a summarization instance, to facilitate the long-context evaluation.

length-control experiments. They lack an objective and comprehensive understanding of the model's capability across different dimensions of long sequences.

In this study, we aim to maximize the diversity of the constructed tasks and analyze the long-context capabilities of LLMs from a user's practical perspective. We discovered that when processing instructions based on long sequences, the essential components for task completion can be classified as single-span, multiple-span, or global, based on relevance. Building on this and considering how to locate these information, we categorize long-context understanding into five distinct abilities and introduce an automated method to transform short-sequence tasks into a comprehensive long-sequence scenario encompassing all these capabilities. As a result, M⁴LE is proposed, a multi-ability, multi-range, multi-task and multi-domain long-context evaluation benchmark for evaluating LLMs' ability in handling long inputs (Figure 1).

- Multi-ability: M⁴LE includes tasks with five different types of understanding abilities, determined by whether single or multiple parts of the ongoing context are relevant to the current tasks and whether explicit or semantic hints are used in the question.

- Multi-range: Each task in M⁴LE consists of samples with variable lengths, from 1K to 8K words, divided evenly into five buckets to measure the effect of length on model performance.

- Multi-task: M⁴LE encompasses 36 datasets covering 11 task types, including original tasks such as classification and summarization, and their combination for more complex scenarios.

- Multi-domain: M⁴LE spans a wide variety of domains, including Wikipedia, academic, news, E-Commerce, etc., prompting diversity and comprehensiveness.

Table 1 compares M⁴LE with the existing long-context benchmarks. M⁴LE targets on comprehensively evaluating LLMs' long-context understanding capabilities across different abilities and length ranges, rather than simply assessing on naturally long input tasks. Therefore, the tasks in M⁴LE are constructed from both existing long-context datasets and short-context datasets widely used in the NLP community, where short instances can be aggregated into long-context ones with designed procedure covering different abilities with varied instructions. Our approach is able to extend existing datasets to arbitrary context lengths.

| Benchmarks | SCROLLS | ZeroSCROLLS | L-Eval | LongBench | M⁴LE |
|---|---|---|---|---|---|
| #Tasks | 3 | 4 | 4 | 6 | 11 |
| #Datasets | 7 | 10 | 18 | 21 | 36 |
| #Domains | 7 | 9 | 10 | 10 | 12 |
| Languages | en | en | en | en, zh | en, zh |
| Ranges | × | × | × | × | ✓ |
| Abilities | × | × | × | × | ✓ |

Table 1: Comparison with existing benchmarks.

We conducted a systematic evaluation over 11 well-known LLMs, especially those claimed to support long inputs, with M⁴LE. This involves evaluating their long-context understanding ability across

different length ranges and their performance in our proposed five different abilities. We also delve into the factors influencing long-context understanding capability, including LLMs performance under different languages and the positioning of relevant information (Liu et al., 2023). We find that current LLMs still struggle to understand long-context inputs, especially when multiple-span attention is required. While semantic retrieval is considered more complex than explicit, the consistent performance drop in this scope can only be observed on competent models. A more effective fine-tuning approach deserves exploration, as current methods show no significant improvement over simple Neural Tangent Kernel (NTK) aware scaling methods. We also observe that language differences and the positioning of relevant information impact the long-context understanding capabilities.

## 2 RELATED WORK

### 2.1 LONG-CONTEXT MODELLING FOR LLMS

To address length extrapolation challenges in LLMs beyond the training context window, several methodologies have emerged. Position embeddings such as Alibi (Press et al., 2022) and XPos (Sun et al., 2023) have been developed. Alibi employs an exponential decay on the attention matrix to mitigate out-of-distribution positions' influence, while XPos introduces a block-wise causal attention mask. While these techniques require integration during training, alternative approaches enhance existing RoPE-based LLMs (Su et al., 2021), notably LLaMA (Touvron et al., 2023a), LLaMA 2 (Touvron et al., 2023b), and PaLM (Chowdhery et al., 2022). Concurrently, kaiokendev (2023) and Chen et al. (2023) propose extending context length by modifying RoPE through Position Interpolation and subsequent limited data finetuning. Another line of research introduces fine-tuning free approaches (bloc97, 2023; emozilla, 2023; Peng et al., 2023), including NTK-aware and dynamic NTK interpolations.

### 2.2 EXISTING EVALUATION BENCHMARKS FOR LLMS

As LLMs have demonstrated superior performance in a wide range of NLP tasks, comprehensively and effectively evaluating their ability becomes increasingly critical. Many of the research efforts focus on developing benchmarks for specific knowledge types (Hendrycks et al., 2021; Zhong et al., 2023) and specific task families (Chen et al., 2021; Cobbe et al., 2021). For more details, we refer readers to the recent LLMs evaluation survey Chang et al. (2023); Wang et al. (2023). Several preliminary studies have begun to assess the model capability on long context input. Long Range Areana (Tay et al., 2020) verifies the capability of transformer-based models to handle various long sequence inputs, such as languages, vision tokens and symbols. SCROLLS (Shaham et al., 2022) simply collects a set of naturally long NLP benchmarks covering multiple tasks and domains. Recently, ZeroSCROLLS (Shaham et al., 2023), L-Eval (An et al., 2023) and LongBench (Bai et al., 2023) are proposed to evaluate long text modelling capability of LLMs. However, these benchmarks are mainly compiled from a set of existing long NLP benchmarks, thereby suffering from data diversity (i.e., limited evaluation patterns) and data leakage (i.e., LLMs potentially already using these benchmarks for pre-training or alignment). In contrast, M$^4$LE not only constructs evaluation instances from various tasks, domains and length ranges but also covers three types of attention spans, offering a comprehensive evaluation of LLMs' long text capability.

## 3 M$^4$LE

This section introduces the rationale and design principles of the benchmark, as well as the data sources and task construction methodologies. M$^4$LE has been carefully curated to cover a wide range of long-context natural language understanding abilities, task types, domains, and context length ranges, ensuring a thorough reflection of LLM's long-context competencies.

### 3.1 DESIGN PRINCIPLE

Each sample in M$^4$LE is a tuple of ⟨Task description, Context, Instruction, Response⟩. In order to accomplish the instructions, LLMs need to retrieve and identify relevant parts from the long context:

- Those relevant parts could be *single-span*, *multiple-span*, or *global*. A span is a continuous text segment within the long context.

- The retrieval could be based on *explicit* or *semantic* hints in the instruction according to those parts could be explicitly or semantically located.

Accordingly, We break down the understanding ability into five distinctive categories: 1) *explicit single-span* understanding, 2) *semantic single-span* understanding, 3) *explicit multiple-span* understanding, 4) *semantic multiple-span* understanding and 5) *global* context understanding (Figure 1).

We try to maximize the diversity of the constructed tasks in the following aspects:

- Data Source: We select widely-used Chinese and English datasets in NLP which covers a variety of representative task types (e.g., QA, Summarization) and domains (e.g., News, Wiki, Web). In addition, we introduce tasks that integrate multiple task types, like Classification + Retrieval. These newly integrated tasks help measure LLMs' ability of solving more complex tasks.

- Length Level: It is important to reveal how LLMs perform on various lengths of contexts. In our benchmark, we evenly divide samples into buckets according to their context lengths. In addition, in order to alleviate the effects of location of relevant parts in context (Liu et al., 2023), we intentionally construct instances with the relevant paragraphs uniformly distributed in the input context.

## 3.2 DATA COLLECTION

We collect established datasets, both in English and Chinese, to cover a broad range of tasks and domains. We not only select datasets featuring long inputs, but also include datasets with shorter inputs for our customized construction, and at the same time, enriching the domain variety. The short-context datasets can be adapted to longer context using our designed process, which will be introduced in next subsection. Below we describe the datasets selected in the benchmark briefly.

*Question-Answering (QA)*: We include TriviaQA (Joshi et al., 2017), a single-document QA dataset based on web snippets and Wikipedia, with documents extended to 12k words. Additionally, NQ-Open (Lee et al., 2019), HotpotQA (Yang et al., 2018), and DRCD (Shao et al., 2019) are included, all of which are based on Wikipedia articles. We further collect NewsQA (Trischler et al., 2017) and DuoRC (Saha et al., 2018), both in English and constructed from news articles and movie plots. We also add C3 (Sun et al., 2021), a Chinese dataset comprising textbook questions.

*Classification*: We incorporate BIGPATENT (Sharma et al., 2019) which includes long patent documents, and MNDS News (Petukhova & Fachada, 2023) in English and THUCNews (Hu et al., 2019) in Chinese which would be further processed for different abilities. We also utilize a sentiment classification dataset collected from e-commerce platforms (SophonPlus, 2013).

*Summarization*: For English, we include Arxiv, Pubmed (Cohan et al., 2018), BIGPATENT (Sharma et al., 2019), and Booksum (Kryscinski et al., 2022), where the corresponding domains span across academic, medical, patent documents and books. We also introduce shorter summarization datasets enabling extension, such as CNNNews (See et al., 2017) and MNDS News, featuring news articles, and Wikihow (Koupaee & Wang, 2018). For Chinese, we incorporate CNewsum (Wang et al., 2021), CLTS+ (Liu et al., 2022), and News2016 (Xu, 2019), all constructed from long news articles. The LCSTS (Hu et al., 2015) dataset contains shorter news articles, while CEPSUM (Li et al., 2020) comprises product descriptions from e-commerce platforms. We also use NCLS (Zhu et al., 2019) to establish a bilingual task that generates a Chinese summary for a specific English news article.

*Natural Language Inference (NLI)*: We construct two tasks using English and Chinese Wikipedia articles from WikiText-103 (Merity et al., 2016) and Wiki2019zh (Xu, 2019), respectively.

*Translation*: Three translation datasets are included, depending sentence-level translation alignments to form long contexts, including Tedtalks (Qi et al., 2018), OpenSubtitles (Lison & Tiedemann, 2016), and News commentary (Tiedemann, 2012).

*Retrieval*: Lastly, we construct two retrieval tasks from the same datasets used for the NLI task for both languages. Since $M^4LE$ comprises numerous tasks combined with retrieval capability, we do not construct additional standalone retrieval datasets.

## 3.3 TASK CONSTRUCTION

Table 3 provides an overview of the constructed datasets in M⁴LE. The detailed statistics of the datasets used can be found in Appendix A.1. In this subsection, we introduce how we construct the datasets under the categories of five abilities.

Instances for each ability are derived from the data pool collected above. For each dataset, we construct instances with input context lengths in diverse length ranges. To construct an instance of a specific task (described by "Task description") in length range $K$, we sample $N$ original instances randomly from a single source dataset and combine their context paragraphs into a long sequence as "Context", where each paragraph is marked with an explicit identifier at the beginning for indexing. The value of $N$ is calculated by dividing the desired length range $K$ by the median length of the original instances. Then "Instruction" is generated to guide models on what objective to complete, resulting in different abilities to be evaluated. Additionally, each task is supplemented with a one-shot exemplar to demonstrate the desired output. This approach allows us to extend existing datasets with short contexts to arbitrary context lengths. Below are the specific instructions for five abilities.

| Ability | Dataset | Task Type | Language | Domain | Metric | Ave. Len. |
|---|---|---|---|---|---|---|
| Explicit Single | MNDS News | CLS + RET | En | News | Acc | 3805 |
| | THUCNews | CLS + RET | Zh | News | Acc | 3650 |
| | NewsQA | QA + RET | En | News | Acc | 3679 |
| | C3 | QA + RET | Zh | Textbook | Acc | 3797 |
| | WoW | RET | En | Wiki | Acc | 3434 |
| | DRCD | RET | Zh | Wiki | Acc | 3617 |
| | CNNNews | SUM + RET | En | News | Rouge-L | 3754 |
| | CEPSUM | SUM + RET | Zh | E-Commerce | Rouge-L | 4003 |
| | LCSTS | SUM + RET | Zh | News | Rouge-L | 4102 |
| | NCLS | SUM + RET | En,Zh | News | Rouge-L | 3470 |
| Explicit Multiple | MNDS News | CLS + RET | En | News | F1 | 3772 |
| | THUCNews | CLS + RET | Zh | News | F1 | 3721 |
| | MARC | CLS + RET | En,Zh | E-Commerce | F1 | 3543 |
| | Online Shopping | CLS + RET | Zh | E-Commerce | F1 | 3714 |
| Semantic Single | WikiText-103 | NLI + RET | En | Wiki | Acc | 3278 |
| | Wiki2019zh | NLI + RET | Zh | Wiki | Acc | 3723 |
| | DuoRC | QA | En | Movie | Acc | 3572 |
| | NQ-Open | QA | En | Wiki | Acc | 3128 |
| | DuReader | QA | Zh | Web | Acc | 3261 |
| | DRCD | QA | Zh | Wiki | Acc | 3300 |
| | WikiHow | SUM + RET | En | WikiHow | Rouge-L | 3514 |
| | News2016 | SUM + RET | Zh | News | Rouge-L | 3785 |
| | TedTalks | TRAN + RET | En,Zh | TedTalks | BLEU | 2956 |
| Semantic Multiple | MNDS News | CLS + CNT | En | News | Acc | 3791 |
| | THUCNews | CLS + CNT | Zh | News | Acc | 3699 |
| | HotpotQA | QA | En | Wiki | Acc | 1060 |
| Global | BIGPATENT | CLS | En | Patent | Acc | 3407 |
| | TriviaQA | QA | En | Web | Acc | 3329 |
| | Arixv | SUM | En | Academic | Rouge-L | 3748 |
| | BIGPATENT | SUM | En | Patent | Rouge-L | 3293 |
| | Pubmed | SUM | En | Medical | Rouge-L | 3678 |
| | Booksum | SUM | En | Book | Rouge-L | 2643 |
| | CNewsum | SUM | Zh | News | Rouge-L | 1883 |
| | CLTS+ | SUM | Zh | News | Rouge-L | 3158 |
| | OpenSubtitles | TRAN | En,Zh | Movie | BLEU | 2048 |
| | News Commentary | TRAN | En,Zh | News | BLEU | 3585 |

Table 2: The overview of the evaluated tasks in M⁴LE, categorized by abilities. CLS, QA, RET, SUM, TRAN, and CNT denote classification, question-answering, retrieval, summarization, translation, and counting respectively. Acc in metric stands for accuracy.

**Explicit Single-Span Understanding.** Instructions for tasks within this scope should direct models to complete the task based on a specific paragraph, with explicit hints to be located. For instance, in a question-answering task, the model might be asked to answer a question based on paragraph II. This approach has been used to construct ten unique datasets covering a wide range of task types and domains for the ability. Consequently, the task types are a fusion of retrieval and their original task, such as classification, which is labeled as "CLS + RET".

**Semantic Single-Span Understanding.** Analogous to explicit single-span understanding, the instructions for the tasks long to this ability instruct models to complete tasks based on a designated paragraph. Rather than using explicit identifiers, we provide hints about the paragraph and models are tasked with retrieving it based on semantic information. For example, in a translation task, the model might be prompted to translate a paragraph associated with sports. Tasks within this ability are designed to introduce increased complexity and challenges since semantic-level retrieval necessitates the model to understand all paragraphs to pinpoint the right one. We have constructed nine distinct datasets aligned with this ability.

**Explicit Multiple-Span Understanding.** We add further complexities to the tasks within this ability. Specifically, models are tasked with handling assignments related to multiple, disjoint paragraphs within the lengthy input context. This could necessitate addressing several original instances, for example, summarizing the first and the third paragraphs. Despite these complexities, the instructions for this ability continue to utilize explicit hints. We have constructed four distinct datasets to align with this ability.

**Semantic Multiple-Span Understanding.** We replace the explicit hints in explicit multiple-span understanding with semantic ones, resulting in the instructions for tasks in this scope. We've developed three distinct datasets of high complexity in line with this. Within this ability, we've incorporated counting tasks (labelled as "CNT"), which demand the counting of relevant paragraphs. Such tasks pose a challenge since counting is not an innate function of language models.

**Global Context Understanding.** Finally, we present tasks in global context understanding, which is a special case within our construction process. When the original instances have sufficiently extensive context, such that the target length range $K$ can be attained with $N = 1$, we directly employ them for the associated tasks, indicating that the entire context is relevant to the task completion and global understanding is required. Within this category, we have included ten different datasets.

## 4 EXPERIMENTS

### 4.1 MODELS

We introduce the five families of LLMs evaluated in this study, comprising a total of 11 models.

**LLaMA 2:** It is a family of LLMs that support a maximum 4k input length (Touvron et al., 2023b). These models use rotary positional embeddings (RoPE) (Su et al., 2021). LLaMA 2 has 7B, 13B and 70B variant. We focus on its 7B and 13B models in this section. We also include their aligned versions: LLaMA2-7B-Chat and LLaMA2-13B-Chat.

**Vicuna:** We employ Vicuna-7B-v1.5-16K and Vicuna-13B-v1.5-16K (Chiang et al., 2023), fine-tuned based on the LLaMA2 models with 125k conversational data, collected from ShareGPT with context length up to 16K tokens using linear positional interpolation (Chen et al., 2023).

**LongChat:** We leverage LongChat-7B-v1.5-32K and LongChat-13B-16K (Li et al., 2023), fine-tuned on 80K and 18K conversations respectively, with context lengths up to 32K and 16K tokens, respectively. They utilize linear positional interpolation.

**ChatGLM2:** ChatGLM2-6B and ChatGLM2-6B-32K are based on the GLM (Du et al., 2022) models. Similar to LLaMA2, ChatGLM2 leverage RoPE. Both models are further refined on 8K and 32K input data, respectively, using linear positional interpolation.

**GPT-3.5-Turbo:** It is a closed-source language model developed based on InstructGPT (Ouyang et al., 2022). Analogous to LLaMA 2, it is fine-tuned with instruction data and refined by RLHF. We use the GPT-3.5-Turbo-16K variant [2], which supports a 16K context length.

---

[2]We use the GPT-3.5-Turbo-16K-0613 api from https://cuhk-api-dev1-apim1.developer.azure-api.net.

## 4.2 INFERENCE DETAILS

Apart from the tuples introduced in Section 3.1, we also employ a concise and short in-context example, from the same dataset, to demonstrate the desired output format. Several full examples used in this work can be found in Appendix A.3. The main goal of M$^4$LE is to evaluate the performance variations across different context length buckets and abilities. We did not perform extensive prompt engineering for each task to obtain the optimal performance. Instead, we focus on analysing performance changes of particular LLMs with longer input context.

Since LLaMA 2 models were trained on data within 4k tokens, we used dynamic NTK-aware RoPE scaling (emozilla, 2023; Peng et al., 2023) for context longer than 4k. We used 16 floating points precision during inference. To facilitate fair comparisons across various tasks with different metrics, we normalized the raw performance score $r(M, l)$ (i.e., the performance of LLM $M$ at context length $l$) as follows:

$$\hat{r}(M, l) = \frac{r(M, l)}{r(\text{GPT-3.5-Turbo-16K}, 1000) + r(M, l)}$$

$\hat{r}(M, l)$ provides a measure of how other models perform relative to GPT-3.5-Turbo-16K in the length range bucket of 0-1000 tokens, and how their performance deteriorates with longer input.

## 4.3 RESULTS

Figure 2 illustrates the changes in normalized average scores for various evaluated models as context lengths extend, and Figure 3 depicts their ability in the context length range of 0-1000, 1000-4000, and 4000-8000 (the full results for each task can be found in Appendix A.4). Based on the figures, several key observations emerge:

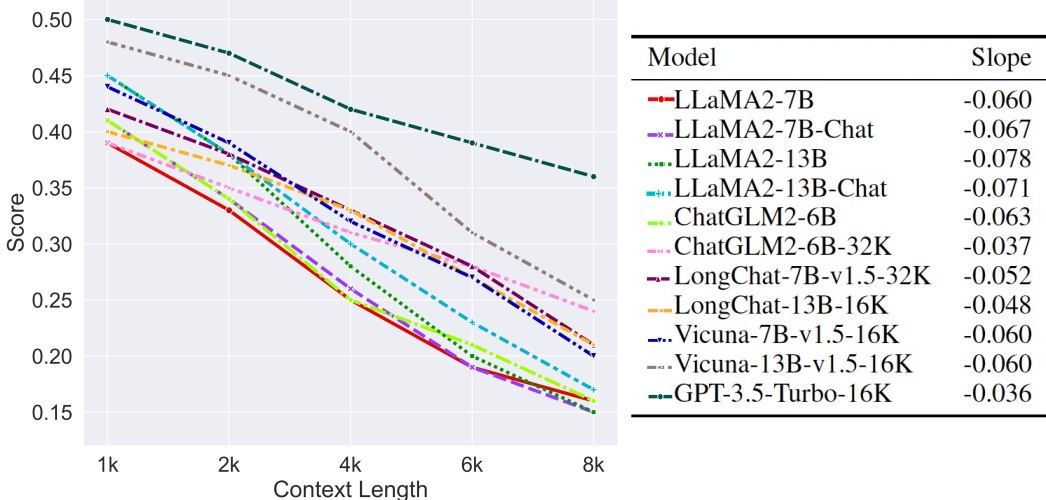

Figure 2: The normalized scores of various models in different context lengths (left), accompanied by the slopes of the corresponding best-fit lines (right). The performance of all models deteriorates with increasing context length.

**The performance of all models significantly deteriorates with increasing context lengths.** This trend is expected, given that a longer context might necessitate more sophisticated modelling capabilities. It suggests that these LLMs struggle with understanding extensive context. The performance gap between GPT-3.5 and most open-source models widens as context length increases. This is largely because open-source models tend to exhibit a steeper decline, particularly when the context length exceeds 4k. For example, Vicuna-13B-v1.5-16K achieves competitive performance, compared to GPT-3.5-Turbo-16K, in the 0-4K length range, but its performance drops significantly after that. A notable exception is ChatGLM2-6B-32k which achieves similar performance when testing on 6K and 8K instances and is only surpassed by GPT-3.5-Turbo-16K on 8K instances.

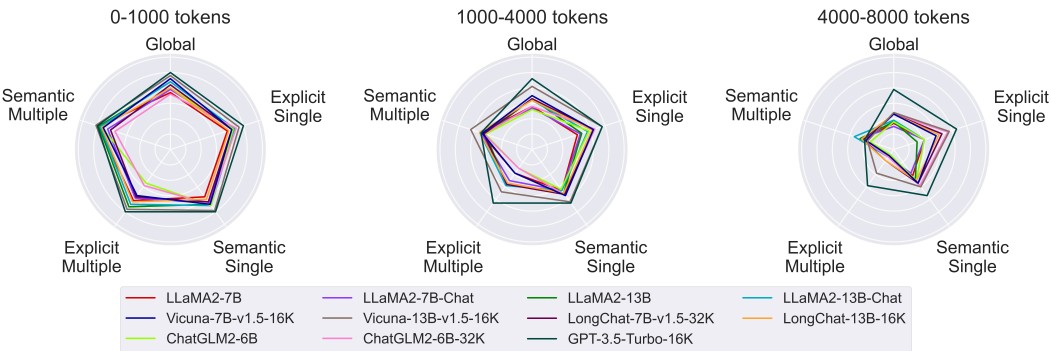

Figure 3: The comparison of abilities of various models in three context length ranges, respectively. It shows that multi-span understanding is more difficult in general. While semantic retrieval appears to be intuitively more challenging, our findings indicate that it is only more demanding for competent models such as GPT-3.5-Turbo-16K at longer lengths.

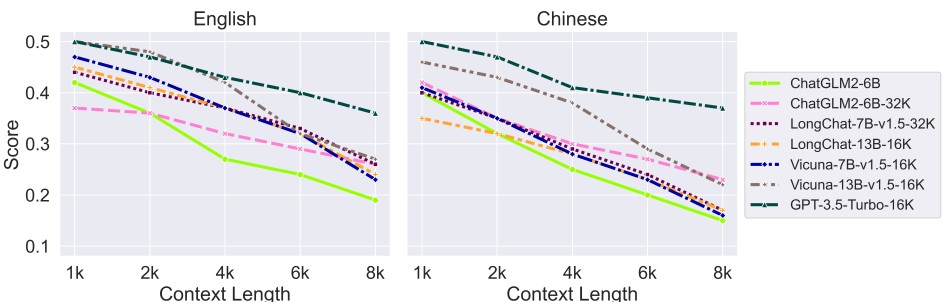

Figure 4: The normalized performance of the models fine-tuned in longer data for English and Chinese tasks, respectively. All models exhibit a similar trend in the decline of performance in both languages, highlighting a universal decline in comprehension as the context length increases, not limited to English.

**Fine-tuning with additional long context data does not offer a significant advantage over simply NTK scaling for understanding long contexts.** Both Vicuna and LongChat models are claimed to support long context as they are directly fine-tuned with longer context data. However, their performance still drops quickly when context length exceeds 4k, with no additional advantage compared to LLaMA2 models, which are trained only on 4k data and merely equipped with NTK scaling method when context length exceeds 4k. This suggests that existing long-context fine-tuning methods contribute minimally to improving long context understanding and a more efficient and effective way to enhance long context understanding ability is needed.

**Multiple-span understanding is more difficult, and semantic retrieval is even harder for competent models.** There is a significant drop in performance on tasks requiring multiple-span attention as context lengthens. This is expected since attending to multiple positions is naturally harder than a single position, and it might require additional ability to distinguish and determine compared to global understanding. Surprisingly, semantic retrieval is only more challenging for GPT-3.5-Turbo-16K, the most competent model in the experiment. We hypothesize that this is because explicit retrieval, looking for relative information by an identifier, is an unnatural task for less competent and generalized LLM. On the contrary, semantic retrieval is more similar to tasks like QA that these models experienced during instruction fine-tuning.

### 4.3.1 ABLATION STUDY

We perform further analysis to understand how models behave in different languages and locations of the supporting document.

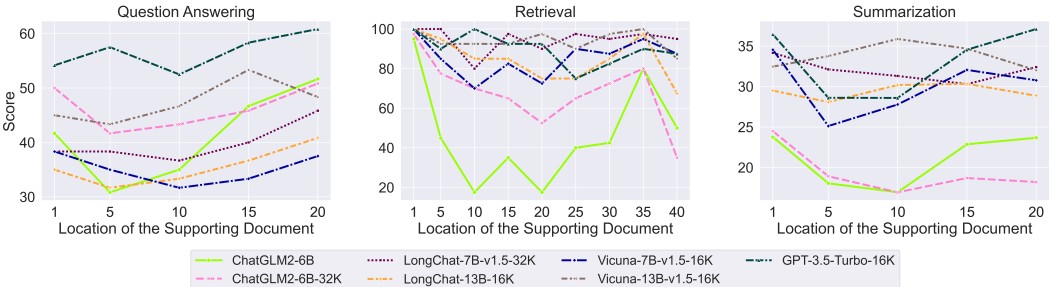

Figure 5: The performance of various models across three tasks, with the supporting document located at different relative positions. It shows higher performance is often obtained when the supporting document is positioned either at the beginning or the end, consistent with Liu et al. (2023).

**Impact of language differences on long-context understanding.** Tasks in different languages may have distinct ability requirements due to the nature of languages and the effects of tokenization. While most models presented in this study are primarily trained on English data, we aim to assess the influence of language differences on the results. In Figure 4, we compare the performance of the top-performing models (namely GPT-3.5-Turbo-16K, ChatGLM2, Vicuna, and LongChat) in both Chinese and English tasks to determine if their long-context understanding abilities differ across languages.

We observe a comparable decline in performance for all models across the two languages. This suggests that the degradation of understanding ability when the context length increases is not unique to English. Furthermore, the diversity of data employed during fine-tuning, as highlighted by Chat-GLM2's emphasis on its bilingual (Chinese and English) proficiency during its tuning process, appears to be a successful strategy in handling bilingual long context input, evidenced by the modest degradation in both languages.

**Lost-in-the-middle exists in other NLP long sequence tasks.** Recently, Liu et al. (2023) find that LLMs tend to ignore the information in the middle of long input context for the task of question answering and retrieval. In this section, following the setup in Liu et al. (2023), we conduct a comprehensive experiment to study the impact of positions of the supporting paragraphs within the context based on our proposed $\texttt{M}^4\texttt{LE}$ benchmark. Specifically, we generate additional instances from the tasks in $\texttt{M}^4\texttt{LE}$, each containing an identical input but with the supporting paragraph placed at different locations. We employ four datasets for question-answering and summarization, and two datasets for retrieval tasks. For the setup details please refer to Appendix A.2.

The average score for each relative position of the supporting document across the three tasks is presented in Figure 5, demonstrating that models typically perform better when the supporting document is positioned either at the beginning or the end of the context, a finding consistent with Liu et al. (2023). Consequently, this suggests that the tendency for LLM to ignore information in the middle of the context is ubiquitous across various languages, models, and tasks. This also shows the potential of $\texttt{M}^4\texttt{LE}$ in discovering interesting and unique LLMs behavior in the long context scenario.

## 5 CONCLUSION

In this paper, we propose a benchmark $\texttt{M}^4\texttt{LE}$ for LLMs assessing their capability of long-context understanding. To establish a benchmark with diverse NLP tasks, rather than just those that are inherently lengthy, we propose a systematic method to convert short NLP task instances into long context inputs, encompassing five distinct abilities. We collect and construct in total of 36 tasks from different sources and domains covering multiple length ranges to maximize the diversity of the tasks in benchmark, with our customized construction methods which enable flexibility to extend arbitrary context lengths. We evaluate 11 well-known LLMs with our benchmark and find that current models struggle to understand long-context inputs and the corresponding performance related to ability types, data used when fine-tuning, and positions of the relevant information.

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

# A APPENDIX

## A.1 DATASETS

This section describes the datasets used in M⁴LE.

| Ability | Dataset | Ave. #words | Max #words | Ave. #sent. | Max #sent. | #instances |
|---|---|---|---|---|---|---|
| Explicit Single | MNDS News | 3805 | 31998 | 488 | 1503 | 1601 |
| | THUCNews | 3650 | 31993 | 205 | 699 | 1801 |
| | NewsQA | 3679 | 31998 | 348 | 1109 | 1812 |
| | C3 | 3797 | 31999 | 289 | 864 | 1842 |
| | WoW | 3434 | 30162 | 402 | 1225 | 1816 |
| | DRCD | 3617 | 29697 | 171 | 518 | 1817 |
| | CNNNews | 3754 | 31988 | 579 | 1763 | 1820 |
| | CEPSUM | 4003 | 32000 | 27 | 114 | 1820 |
| | LCSTS | 4102 | 31995 | 69 | 317 | 1840 |
| | NCLS | 3470 | 31553 | 502 | 1795 | 1811 |
| Explicit Multiple | MNDS News | 3772 | 31995 | 498 | 1561 | 1800 |
| | THUCNews | 3721 | 31994 | 205 | 700 | 1800 |
| | MARC | 3543 | 31044 | 306 | 982 | 1800 |
| | Online Shopping | 3714 | 30691 | 274 | 1026 | 1800 |
| Semantic Single | WikiText-103 | 3278 | 31874 | 421 | 1302 | 1811 |
| | Wiki2019zh | 3723 | 31999 | 418 | 1310 | 1805 |
| | DuoRC | 3572 | 31985 | 451 | 1502 | 1805 |
| | NQ-Open | 3128 | 31993 | 440 | 1624 | 1819 |
| | DuReader | 3261 | 11965 | 67 | 546 | 1200 |
| | DRCD | 3300 | 29858 | 209 | 692 | 1817 |
| | WikiHow | 3514 | 11997 | 282 | 745 | 1811 |
| | News2016 | 3785 | 31992 | 220 | 667 | 1814 |
| | TedTalks | 2956 | 30575 | 597 | 1801 | 1810 |
| Semantic Multiple | MNDS News | 3791 | 31998 | 472 | 1519 | 1813 |
| | THUCNews | 3699 | 31998 | 208 | 695 | 1808 |
| | HotpotQA | 1060 | 1691 | 51 | 86 | 400 |
| Global | BIGPATENT (CLS) | 3407 | 31789 | 324 | 6281 | 1713 |
| | TriviaQA | 3329 | 31678 | 346 | 1946 | 1643 |
| | Arixv | 3748 | 23575 | 295 | 1215 | 1396 |
| | BIGPATENT (SUM) | 3293 | 31818 | 323 | 6272 | 1716 |
| | Pubmed | 3678 | 11974 | 151 | 432 | 1046 |
| | Booksum | 2643 | 31984 | 511 | 4875 | 1800 |
| | CNewsum | 1883 | 5069 | 41 | 167 | 690 |
| | CLTS+ | 3158 | 7462 | 69 | 317 | 1032 |
| | OpenSubtitles | 2048 | 32000 | 597 | 2333 | 1800 |
| | News Commentary | 3585 | 29860 | 339 | 1025 | 1805 |

Table 3: The detailed statistics of different tasks. We report the average (Ave.) and maximum number of words and sentences in each task.

### A.1.1 MNDS NEWS

MNDS News (Petukhova & Fachada, 2023) is an English hierarchical news category classification dataset comprising 10,917 news articles from 260 sources. We only use the 17 first-level categories as the labels for this study. For multiple retrieval tasks, we randomly sample a class label that appears in the instance.

### A.1.2 THUCNEWS

THUCNews (Hu et al., 2019) is a Chinese classification dataset containing 74 million news articles from Sina, with each article belonging to one of the ten categories. We filter out the articles with the number of words less than 20. The multiple retrieval task is built similarly to MNDS News.

### A.1.3   MARC

MARC (Keung et al., 2020) is a dataset for the bilingual (English and Chinese) setting. It contains multilingual Amazon reviews with star ratings from 1 to 5, where 5 is the best. We use 1-star and 5-star reviews for negative and positive reviews respectively, and ask models to return all positive reviews.

### A.1.4   ONLINE SHOPPING

Online Shopping (SophonPlus, 2013) is a Chinese sentiment dataset containing 60K product reviews from Chinese e-commerce platforms. Each review is marked as positive or negative.

### A.1.5   BIGPATENT

BIGPATENT (Sharma et al., 2019) consists of 1.3 million records of U.S. BIGPATENT documents across nine technological areas. The abstract of the document is used as the golden document summary.

### A.1.6   CEPSUM

CEPSUM (Li et al., 2020) is a dataset containing product descriptions and summary pairs collected from a popular Chinese e-commerce platform. We removed instances with less than 60 words in the product description.

### A.1.7   CNNNEWS

CNNNews (See et al., 2017) contains English online news articles from CNN, where each of it is paired with a multi-sentence summary. We preprocess the data using the script from See et al. (2017) and select the instances with at least 30 words in the article.

### A.1.8   LCSTS

LCSTS (Hu et al., 2015) is a Chinese summarization dataset consisting of over 2 million posts and short summary pairs collected from the Chinese microblogging website Sina Weibo. We use part two of the data, which consists of 10,666 (text, summary) pairs with a human-labeled score to indicate the relevance between the post and the summary. The score ranges from 1 to 5, where 5 indicates the most relevant. We select only the samples with a score of 5 in the relevance score.

### A.1.9   NCLS

NCLS (Zhu et al., 2019) is a cross-lingual summarization dataset consisting of pairs of articles in one language and summaries in another language (Chinese or English), constructed from the CNNNews and LCSTS datasets.

### A.1.10   WIKIHOW

WikiHow (Koupaee & Wang, 2018) comprises 230,000 English articles that describe a procedural task along with corresponding summaries. Each article has a title that starts with "How to". The procedures described in the article are separated into multiple steps, where each step corresponds to a paragraph. Each paragraph has a short summary. These summaries are concatenated to form the summary of the article. We remove instances with articles that have less than 60 words.

### A.1.11   NEWS2016

News2016 (Xu, 2019), encompassing over 2 million Chinese news articles. Each article contains a title and keywords. The title is used as the golden summary of the news article. We remove instances with the number of words less than 200 and more than 800.

### A.1.12 ARXIV

Arxiv (Cohan et al., 2018) consists of 215K academic papers from arXiv.org. The abstracts of the papers are used as the golden summary.

### A.1.13 BOOKSUM

Booksum (Kryscinski et al., 2022), which includes 405 English books including plays, short stories, and novels with human-written summaries for each chapter. We combine the consecutive chapters and the corresponding summaries to construct instances for any context length range.

### A.1.14 CNEWSUM

CNewsum (Wang et al., 2021) contains 304,307 Chinese news articles from different press publishers with human-written summaries.

### A.1.15 CLTS+

CLTS+ (Liu et al., 2022) is an improved Chinese new articles summarization dataset based on CLTS (Liu et al., 2020). CLTS contains more than 180,000 Chinese long articles with human-written summaries. CLTS+ utilize back translation to enhance the abstractiveness of the summaries.

### A.1.16 NEWSQA

NewsQA (Trischler et al., 2017) is an English QA dataset based on 12,744 news articles from CNN. Crowdsourced workers are recruited to generate 119,633 questions and answers.

### A.1.17 C3

C3 (Sun et al., 2021) is a Chinese textbook-based machine comprehension dataset. The questions are multiple-choice questions collected from exams for second-language Chinese learners.

### A.1.18 DUORC

DuoRC (Saha et al., 2018) is an English question-answer dataset based on 7680 movie plots collected from IMDb and Wikipedia. Crowdsourced workers are hired to create 186,089 unique question-answer pairs.

### A.1.19 NQ-OPEN

NaturalQuestions-Open (NQ-Open) (Lee et al., 2019) is an open-domain question answering dataset based on Wikipedia documents. The questions are collected from Google Search queries. We directly use the processed version from Liu et al. (2023).

### A.1.20 DUREADER

DuReader (He et al., 2018) is an open-domain Chinese machine reading comprehension dataset, consisting of 200K questions collected from Baidu Search.

## A.2 EXPERIMENT DETAILS FOR LOST-IN-THE-MIDDLE

For the experiment in Figure 5, which explores the effects of the positions of the relevant paragraphs, we additionally construct the following instances:

In the QA task, 100 instances, each comprising 20 paragraphs, are generated from NQ-Open and DuoRC for English, and from DRCD and C3 for Chinese. Similarly, for the summarization task, we generate 100 instances each from WikiHow and CNNNews for English and News2016, and LCSTS for Chinese. For the retrieval task, we formulate 200 instances each using WoW for English and DRCD for Chinese. The supporting paragraph will be evenly placed at different locations.

A.3   PROMPTS

In this section, we describe the prompts used in M$^4$LE. The prompt begins with the task definition, followed by the in-context example and the testing instance. Below we show the prompt examples used for each of the five abilities. Other tasks' prompts are constructed similarly.

---

You are given multiple news articles below. Each of them belongs to one of the following categories:
1. crime, law and justice
2. arts, culture, entertainment and media
3. economy, business and finance
4. disaster, accident and emergency incident
5. environment
6. education
7. health
8. human interest
9. lifestyle and leisure
10. politics
11. labour
12. religion and belief
13. science and technology
14. society
15. sport
16. conflict, war and peace
17. weather
You will be asked to return the category of a news article I specified at the end.
Article AD3258: Rarely do the worlds of art and science intersect, but they did with famed Dutch artist Escher.
Even if you do not recognize his name, it is likely you have seen his work without knowing it.
One of the largest collections of his work is now on display in the US.
Article D55E47: On Sunday, NBC's Meet The Press will air an interview with President Donald Trump, conducted by the network's political director, Chuck Todd…
Article 5675E9: The full extent of the ferry disaster in the Iraqi city of Mosul is becoming clearer…
Question: What is the category of article AD3258?
Answer: arts, culture, entertainment and media

Article 11BD15: Read the full article by Catherine Frompovitch at NaturalBlaze
Abstract
The human cytochrome P450 (CYP) superfamily comprises 57 genes. These genes code for enzymes that can have a role in: metabolism of drugs, foreign chemicals, arachidonic acid and eicosanoids; cholesterol metabolism and bile-acid biosynthesis; steroid synthesis and metabolism; vitamin D(3) synthesis and metabolism; retinoic acid hydroxylation;…
Article 92FF60: Undoubtedly, this latest flooding crisis in Iran reveals the highly vicious nature of the current U.S. Administration with regards to the application of collective punishment of a target nation…
….
Question: What is the category of article 11BD15?
Answer:

---

Figure 6: An example prompt for the explicit single retrieval task based on MNDS.

Below are some articles from wikihow. I will ask you to summarize a particular article at the end.
Article 1: It's style that will leave you looking classy and feminine. This is a twist on the classic — messier and more mermaid-like. This more obscure braid requires skill, but results in an interesting look. It's cute, classic, and easy to do. It looks bit medieval and very eye-catching. It's ideal for weddings or other elegant occasions.
Article 2: It's a green app that contains a white phone icon inside a white text bubble. It's at the top-center of the screen. Select the chat with the attachment you wish to download. Select the attachment you wish to download. It's in the upper-right corner of the screen. The attachment has been saved to your Android device.
Question: Summarize the article related to "How to Style Very Long Hair" using a few instructive sentences.
Summary: Do a French braid. Make an intricate fishtail braid. Try a Dutch braid. Do a triple braid. Make a crazy braid. Do a cascading waterfall braid.

Article 1: Make sure that the shoe is the appropriate length and width for your child's foot. If a shoe squeezes a child's foot too much, it can cause the child to have foot conditions such as blisters and calluses. Remember that your child's foot will grow at a rapid pace and that he may need to be fitted every few months for a new size. So, if your child takes off his shoe and you notice that there are red marks on your child's foot, it may be time to take your child in for a new fitting and buy him a new shoe…
Article 2: Learning and studying shouldn't be stressful. Being stressed out can actually make it harder to learn and remember things. Think about the reasons why you're stressed out and try to resolve those reasons (remove them from your life).For example, if you get stressed out about assignments because you leave them to the last minute to finish, create yourself a study schedule. Build enough time into the study schedule so that you finish your assignments well enough in advance of the due dates to eliminate any of the stress you were feeling. If the grades you're receiving aren't that great it can be easy to let negativity take over…
Question: Summarize the article related to "How to Fit Your Kid for Shoes" using a few instructive sentences.
Summary:

Figure 7: An example prompt for the semantic single retrieval task based on Wikihow.

You are given multiple news articles below where each of them belongs to one of the 17 categories. Each article is prefixed with a article id. You will be asked to return the article ids of all articles belong to a particular category.
The article id is 0A1A04. Rarely do the worlds of art and science intersect, but they did with famed Dutch artist Escher. Even if you do not recognize his name, it is likely you have seen his work without knowing it. One of the largest collections of his work is now on display in the US.
The article id is 95A4BF. On Sunday, NBC's Meet The Press will air an interview with President Donald Trump, conducted by the network's political director, Chuck Todd. While Todd's interviews with 2020 Democratic contenders have consisted largely of challenges from the left interspersed with the odd softball, Trump is unlikely to receive the same friendly treatment.
The article id is A7D6BE. The full extent of the ferry disaster in the Iraqi city of Mosul is becoming clearer. Civil Defence says the number of dead is now at least 120, while 100 people are still missing.
Iraq's Prime Minister Adel Abdul Mahdi is formally requesting a local governor be sacked over the incident.
Question: Provide me the article id of all the news articles related to 'arts, culture, entertainment and media.'.
Answer: 0A1A04, 95A4BF.

The article id AE8707. CNN contributor Ana Navarro accused President Donald Trump of being the "enemy" of conservative principles and, indeed, the "American presidency" altogether Friday on CNN…
The article id 5BC439. A widely-anticipated exchange of prisoners between Russia and Ukraine is under way, according to reports. Buses from a Moscow prison believed to be carrying Ukrainian prisoners arrived at the capital's Vnukovo airport on Saturday….
The article id 638F02. Dec. 18 (UPI) -- Thousands of nurses in Northern Ireland walked out in a 12-hour labor strike Wednesday, rallying for better pay and greater patient safety. A total of about 15,000 nurses participated in the walkout…
…
Question: Provide me the article id of all the news articles related to 'society'.
Answer:

Figure 8: An example prompt for the explicit multiple retrieval task based on MNDS.

Answer the question based on the given paragraphs. Note that some paragraphs might be irrelevant.
Paragraph 1: Pratia is a genus of flowering plants in the family Campanulaceae, native to Asia, Australia and New Zealand.
Paragraph 2: Sutherlandia is a genus of flowering plants in the family Fabaceae.
Question: Are Sutherlandia and Pratia in the same family?
Answer: no.

Paragraph 1: The Stresa Festival Orchestra is a formation composed by young and talented musicians, coming from renewed european orchestras, calling by Gianandrea Noseda to perform every year some original production for the Stresa Festival.  The debut of the Orchestra, on 26 August 2003 with Mozart' "Don Giovanni", began the project of the concert performances of different operas: "Così fan tutte" (2004), "Le nozze di Figaro" (2005), "The magic flute" (2006), "La clemenza di Tito" (2007), …
Paragraph 2: The Metropolitan City of Messina (Italian: "Città metropolitana di Messina" ) is a metropolitan city in Sicily, Italy.  Its capital is the city of Messina.  It replaced the Province of Messina and comprises the city of Messina and other 107 municipalities ("comuni").  According to Eurostat the FUA of the metropolitan area of Messina has in 2014 277,584 inhabitants.
Paragraph 3: Pompei (] ) is a city and "comune" in the Metropolitan City of Naples in Italy, home of the ancient Roman ruins part of the UNESCO World Heritage Sites.
Paragraph 4: Banca di Credito Popolare S.C.p.A. (BCP) is an Italian cooperative bank based in Torre del Greco, in Metropolitan City of Naples, Campania.  Most of the revenue of the bank came from the Metropolitan City of Naples, which the bank had 44 branches in the metropolitan city.
…
Question: What Metropolitan City was Massimo Giordano born in?
Answer:

Figure 9: An example prompt for the semantic multiple retrieval task based on HotpotQA.

## A.4   MAIN RESULTS

We report the results used for plotting Figure 2 and 3.

| | 1k | 2k | 4k | 6k | 8k |
|---|---|---|---|---|---|
| LLaMA2-7B | 0.39 | 0.33 | 0.25 | 0.19 | 0.15 |
| LLaMA2-7B-Chat | 0.41 | 0.34 | 0.26 | 0.19 | 0.15 |
| LLaMA2-13B | 0.44 | 0.38 | 0.28 | 0.21 | 0.15 |
| LLaMA2-13B-Chat | 0.45 | 0.38 | 0.30 | 0.23 | 0.17 |
| ChatGLM2-6B | 0.41 | 0.34 | 0.25 | 0.21 | 0.16 |
| ChatGLM2-6B-32K | 0.39 | 0.35 | 0.31 | 0.28 | 0.24 |
| LongChat-7B-v1.5-32K | 0.42 | 0.38 | 0.33 | 0.28 | 0.21 |
| LongChat-13B-16K | 0.40 | 0.37 | 0.33 | 0.27 | 0.21 |
| Vicuna-7B-v1.5-16K | 0.44 | 0.39 | 0.32 | 0.27 | 0.20 |
| Vicuna-13B-v1.5-16K | 0.48 | 0.45 | 0.40 | 0.31 | 0.25 |
| GPT-3.5-Turbo-16K | 0.50 | 0.47 | 0.42 | 0.39 | 0.36 |

Table 4: The average normalized performance of different models in various lengths.

| | Explicit Single | Semantic Single | Explicit Multiple | Semantic Multiple | Global |
|---|---|---|---|---|---|
| LLaMA2-7B | 0.41 | 0.39 | 0.37 | 0.49 | 0.38 |
| LLaMA2-7B-Chat | 0.39 | 0.43 | 0.39 | 0.43 | 0.43 |
| LLaMA2-13B | 0.46 | 0.44 | 0.44 | 0.49 | 0.44 |
| LLaMA2-13B-Chat | 0.44 | 0.44 | 0.44 | 0.48 | 0.45 |
| ChatGLM2-6B | 0.27 | 0.43 | 0.40 | 0.47 | 0.45 |
| ChatGLM2-6B-32K | 0.29 | 0.45 | 0.36 | 0.38 | 0.44 |
| LongChat-7B-v1.5-32K | 0.38 | 0.42 | 0.42 | 0.41 | 0.42 |
| LongChat-13B-16K | 0.42 | 0.40 | 0.40 | 0.47 | 0.40 |
| Vicuna-7B-v1.5-16K | 0.37 | 0.42 | 0.46 | 0.46 | 0.44 |
| Vicuna-13B-v1.5-16K | 0.48 | 0.47 | 0.48 | 0.51 | 0.49 |
| GPT-3.5-Turbo-16K | 0.50 | 0.50 | 0.50 | 0.50 | 0.50 |

Table 5: Performance comparison of various models in different abilities over the 0-1000 tokens.

| | Explicit Single | Semantic Single | Explicit Multiple | Semantic Multiple | Global |
|---|---|---|---|---|---|
| LLaMA2-7B | 0.19 | 0.31 | 0.26 | 0.35 | 0.33 |
| LLaMA2-7B-Chat | 0.25 | 0.33 | 0.27 | 0.35 | 0.33 |
| LLaMA2-13B | 0.28 | 0.35 | 0.32 | 0.32 | 0.35 |
| LLaMA2-13B-Chat | 0.29 | 0.38 | 0.32 | 0.36 | 0.33 |
| ChatGLM2-6B | 0.15 | 0.39 | 0.26 | 0.32 | 0.32 |
| ChatGLM2-6B-32K | 0.15 | 0.43 | 0.28 | 0.33 | 0.36 |
| LongChat-7B-v1.5-32K | 0.28 | 0.42 | 0.33 | 0.33 | 0.36 |
| LongChat-13B-16K | 0.27 | 0.41 | 0.32 | 0.36 | 0.35 |
| Vicuna-7B-v1.5-16K | 0.19 | 0.42 | 0.35 | 0.34 | 0.37 |
| Vicuna-13B-v1.5-16K | 0.34 | 0.48 | 0.41 | 0.42 | 0.42 |
| GPT-3.5-Turbo-16K | 0.43 | 0.48 | 0.46 | 0.35 | 0.43 |

Table 6: Performance comparison of various models in different abilities over the 2000-4000 tokens
.

| | Explicit Single | Semantic Single | Explicit Multiple | Semantic Multiple | Global |
|---|---|---|---|---|---|
| LLaMA2-7B | 0.05 | 0.21 | 0.15 | 0.21 | 0.21 |
| LLaMA2-7B-Chat | 0.06 | 0.21 | 0.15 | 0.23 | 0.19 |
| LLaMA2-13B | 0.05 | 0.17 | 0.18 | 0.23 | 0.24 |
| LLaMA2-13B-Chat | 0.06 | 0.21 | 0.19 | 0.27 | 0.25 |
| ChatGLM2-6B | 0.04 | 0.21 | 0.18 | 0.16 | 0.25 |
| ChatGLM2-6B-32K | 0.06 | 0.37 | 0.23 | 0.21 | 0.29 |
| LongChat-7B-v1.5-32K | 0.09 | 0.33 | 0.23 | 0.21 | 0.27 |
| LongChat-13B-16K | 0.09 | 0.31 | 0.23 | 0.22 | 0.25 |
| Vicuna-7B-v1.5-16K | 0.05 | 0.29 | 0.23 | 0.19 | 0.27 |
| Vicuna-13B-v1.5-16K | 0.19 | 0.38 | 0.24 | 0.18 | 0.30 |
| GPT-3.5-Turbo-16K | 0.29 | 0.43 | 0.39 | 0.20 | 0.37 |

Table 7: Performance comparison of various models in different abilities over the 4000-8000 tokens
.

## A.5 PERFORMANCE IN VARIOUS TASK TYPES

Figure 10 shows the performance trend across the six task types. Overall, we can observe consistent results across different task types. In particular, LLaMA2-7B models generally exhibit inferior

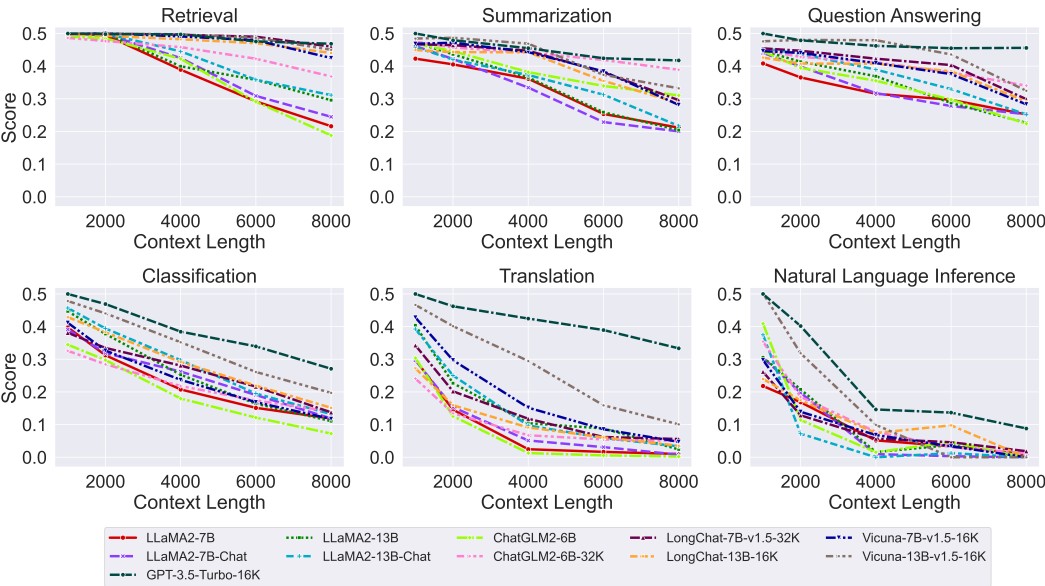

Figure 10: The normalized scores of various models across different task types and context lengths, averaged within each task type.

performance, while Vicuna, LongChat, and ChatGLM2-6B-32K models consistently perform well. Besides, it also shows that retrieval, summarization, and question-answering tasks (the top row in Figure 10) are easier than other tasks.

A significant drop in performance is observed in translation and natural language inference tasks. Natural language inference tasks demand comprehensive understanding to identify the nuances to pick the appropriate continuations, posing substantial challenges. For translation tasks, the model has to generate text that has a similar length to the context, which might be difficult for the model to attend to the correct information. In classification tasks, most models show a linear and gradual drop in performance as context length increases. However, all models have a significant drop in performance for tasks that require counting the number of articles belonging to a certain class. We hypothesize that it might be due to the difficulty of attending and classifying multiple articles at one go since it simply generates a single number in response.

For question-answering and summarization tasks, we find that most models show minimal degradation in performance, with exceptions being the LLaMA2-7B and LLaMA2-7B-Chat in summarization. For retrieval tasks, GPT-3.5, Vicuna, and LongChat models show negligible change in performance. unlike the LLaMA2 models, potentially highlighting the limitations of NTK scaling despite working well in general.

## A.6 TASK RESULTS

| | 1k | 2k | 4k | 6k | 8k |
|---|---|---|---|---|---|
| LLaMA2-7B | 54.00 | 50.75 | 34.48 | 32.37 | 23.08 |
| LLaMA2-7B-Chat | 64.50 | 62.19 | 40.89 | 18.84 | 16.83 |
| LLaMA2-13B | 58.00 | 55.22 | 42.36 | 31.40 | 24.37 |
| LLaMA2-13B-Chat | 64.00 | 62.19 | 44.83 | 36.23 | 25.32 |
| ChatGLM2-6B | 49.00 | 37.81 | 31.53 | 23.67 | 16.83 |
| ChatGLM2-6B-32K | 46.50 | 46.27 | 36.95 | 28.99 | 26.10 |
| LongChat-7B-v1.5-32K | 59.50 | 57.21 | 49.75 | 47.34 | 37.50 |
| LongChat-13B-16K | 59.00 | 52.74 | 49.75 | 48.31 | 24.39 |
| Vicuna-7B-v1.5-16K | 61.00 | 59.70 | 50.74 | 44.93 | 31.73 |
| Vicuna-13B-v1.5-16K | 65.00 | 59.20 | 54.19 | 51.21 | 24.39 |
| GPT-3.5-Turbo-16K | 62.00 | 59.70 | 55.17 | 51.69 | 46.63 |

Table 8: NQ-Open (QA)

| | 1k | 2k | 4k | 6k | 8k |
|---|---|---|---|---|---|
| LLaMA2-7B | 78.00 | 71.00 | 45.00 | 47.26 | 33.50 |
| LLaMA2-7B-Chat | 83.00 | 76.00 | 43.00 | 43.28 | 34.52 |
| LLaMA2-13B | 82.00 | 81.00 | 74.00 | 50.40 | 42.70 |
| LLaMA2-13B-Chat | 88.00 | 83.00 | 77.50 | 51.84 | 45.32 |
| ChatGLM2-6B | 79.00 | 74.00 | 67.50 | 56.22 | 41.00 |
| ChatGLM2-6B-32K | 81.50 | 74.50 | 69.50 | 72.14 | 67.00 |
| LongChat-7B-v1.5-32K | 81.00 | 77.50 | 70.50 | 77.61 | 72.00 |
| LongChat-13B-16K | 66.00 | 60.00 | 51.50 | 54.73 | 47.45 |
| Vicuna-7B-v1.5-16K | 85.00 | 84.50 | 80.50 | 83.58 | 73.50 |
| Vicuna-13B-v1.5-16K | 88.50 | 91.50 | 84.50 | 82.59 | 74.32 |
| GPT-3.5-Turbo-16K | 89.00 | 90.50 | 85.50 | 86.57 | 79.50 |

Table 9: DRCD (QA)

| | 1k | 2k | 4k | 6k | 8k |
|---|---|---|---|---|---|
| LLaMA2-7B | 98.50 | 95.52 | 64.00 | 35.91 | 23.12 |
| LLaMA2-7B-Chat | 97.50 | 99.50 | 82.50 | 46.38 | 32.02 |
| LLaMA2-13B | 98.50 | 99.50 | 83.00 | 62.57 | 47.92 |
| LLaMA2-13B-Chat | 97.50 | 99.00 | 84.00 | 58.45 | 48.92 |
| ChatGLM2-6B | 97.00 | 93.03 | 65.00 | 32.37 | 15.87 |
| ChatGLM2-6B-32K | 93.50 | 91.54 | 86.50 | 74.88 | 54.33 |
| LongChat-7B-v1.5-32K | 98.50 | 99.50 | 97.50 | 97.10 | 76.92 |
| LongChat-13B-16K | 98.00 | 99.00 | 94.00 | 90.82 | 74.93 |
| Vicuna-7B-v1.5-16K | 98.50 | 99.50 | 94.50 | 91.79 | 64.42 |
| Vicuna-13B-v1.5-16K | 98.50 | 99.00 | 98.50 | 92.27 | 76.92 |
| GPT-3.5-Turbo-16K | 98.50 | 98.51 | 97.50 | 90.82 | 87.98 |

Table 10: WoW (RET)

|  | 1k | 2k | 4k | 6k | 8k |
|---|---|---|---|---|---|
| LLaMA2-7B | 99.00 | 99.50 | 62.50 | 46.43 | 31.96 |
| LLaMA2-7B-Chat | 100.00 | 97.51 | 65.00 | 42.37 | 32.39 |
| LLaMA2-13B | 99.50 | 99.50 | 52.00 | 48.70 | 35.88 |
| LLaMA2-13B-Chat | 98.50 | 99.50 | 75.50 | 52.56 | 41.02 |
| ChatGLM2-6B | 94.00 | 94.03 | 81.00 | 50.24 | 31.10 |
| ChatGLM2-6B-32K | 94.50 | 89.55 | 81.50 | 70.53 | 61.72 |
| LongChat-7B-v1.5-32K | 100.00 | 99.00 | 98.00 | 93.72 | 92.82 |
| LongChat-13B-16K | 98.00 | 94.03 | 91.00 | 85.51 | 81.49 |
| Vicuna-7B-v1.5-16K | 99.00 | 99.50 | 97.00 | 90.82 | 83.35 |
| Vicuna-13B-v1.5-16K | 100.00 | 99.50 | 98.00 | 96.14 | 85.79 |
| GPT-3.5-Turbo-16K | 100.00 | 98.51 | 99.00 | 89.37 | 87.08 |

Table 11: DRCD (RET)

|  | 1k | 2k | 4k | 6k | 8k |
|---|---|---|---|---|---|
| LLaMA2-7B | 11.62 | 12.96 | 11.72 | 8.46 | 3.57 |
| LLaMA2-7B-Chat | 14.19 | 14.68 | 16.79 | 8.40 | 4.59 |
| LLaMA2-13B | 13.51 | 13.24 | 12.34 | 9.38 | 5.86 |
| LLaMA2-13B-Chat | 13.47 | 13.56 | 13.96 | 11.46 | 5.93 |
| ChatGLM2-6B | 12.88 | 13.22 | 12.63 | 10.32 | 6.81 |
| ChatGLM2-6B-32K | 13.71 | 14.28 | 14.24 | 12.39 | 8.00 |
| LongChat-7B-v1.5-32K | 14.14 | 14.80 | 14.39 | 10.81 | 8.11 |
| LongChat-13B-16K | 11.94 | 13.42 | 13.48 | 8.75 | 7.15 |
| Vicuna-7B-v1.5-16K | 15.14 | 15.35 | 15.29 | 11.63 | 6.47 |
| Vicuna-13B-v1.5-16K | 14.28 | 14.81 | 14.07 | 8.37 | 6.92 |
| GPT-3.5-Turbo-16K | 18.00 | 16.98 | 15.65 | 12.18 | 10.86 |

Table 12: Booksum (SUM)

|  | 1k | 2k | 4k | 6k | 8k |
|---|---|---|---|---|---|
| LLaMA2-7B | 87.50 | 88.50 | 84.00 | 73.00 | 65.00 |
| LLaMA2-7B-Chat | 86.00 | 86.50 | 76.00 | 64.00 | 63.50 |
| LLaMA2-13B | 90.50 | 92.00 | 82.00 | 75.50 | 61.00 |
| LLaMA2-13B-Chat | 90.50 | 89.00 | 80.50 | 73.00 | 66.00 |
| ChatGLM2-6B | 78.50 | 66.00 | 52.00 | 54.00 | 32.50 |
| ChatGLM2-6B-32K | 77.50 | 76.00 | 61.50 | 58.50 | 45.50 |
| LongChat-7B-v1.5-32K | 87.50 | 84.50 | 80.00 | 75.50 | 68.50 |
| LongChat-13B-16K | 85.00 | 86.50 | 75.00 | 75.50 | 50.00 |
| Vicuna-7B-v1.5-16K | 91.00 | 87.50 | 84.50 | 78.50 | 56.50 |
| Vicuna-13B-v1.5-16K | 88.50 | 85.00 | 80.00 | 77.00 | 50.00 |
| GPT-3.5-Turbo-16K | 89.50 | 83.00 | 82.00 | 77.00 | 73.50 |

Table 13: TriviaQA (QA)

| 1k | 2k | 4k |
|---|---|---|
| LLaMA2-7B | 47.50 | 36.50 |
| LLaMA2-7B-Chat | 44.50 | 42.00 |
| LLaMA2-13B | 52.50 | 39.50 |
| LLaMA2-13B-Chat | 51.50 | 41.00 |
| ChatGLM2-6B | 43.50 | 31.50 |
| ChatGLM2-6B-32K | 41.50 | 35.00 |
| LongChat-7B-v1.5-32K | 49.50 | 40.50 |
| LongChat-13B-16K | 55.00 | 43.50 |
| Vicuna-7B-v1.5-16K | 50.00 | 44.50 |
| Vicuna-13B-v1.5-16K | 56.00 | 52.00 |
| GPT-3.5-Turbo-16K | 55.00 | 41.50 |

Table 14: HotpotQA (QA)

| | 1k | 2k | 4k | 6k | 8k |
|---|---|---|---|---|---|
| LLaMA2-7B | 10.03 | 8.71 | 8.08 | 7.55 | 4.69 |
| LLaMA2-7B-Chat | 21.91 | 16.95 | 13.00 | 9.62 | 5.52 |
| LLaMA2-13B | 19.99 | 15.91 | 16.73 | 12.07 | 6.29 |
| LLaMA2-13B-Chat | 19.19 | 13.48 | 11.73 | 12.38 | 5.48 |
| ChatGLM2-6B | 16.82 | 14.48 | 11.78 | 10.35 | 7.01 |
| ChatGLM2-6B-32K | 20.76 | 20.18 | 18.22 | 14.43 | 14.97 |
| LongChat-7B-v1.5-32K | 22.18 | 23.60 | 23.81 | 14.81 | 18.46 |
| LongChat-13B-16K | 24.11 | 25.46 | 22.97 | 16.20 | 13.20 |
| Vicuna-7B-v1.5-16K | 23.59 | 23.39 | 21.28 | 19.06 | 8.22 |
| Vicuna-13B-v1.5-16K | 24.22 | 23.99 | 18.65 | 12.49 | 10.83 |
| GPT-3.5-Turbo-16K | 21.64 | 21.20 | 20.33 | 17.66 | 14.84 |

Table 15: Arxiv (SUM)

| | 1k | 2k | 4k | 6k | 8k |
|---|---|---|---|---|---|
| LLaMA2-7B | 24.17 | 23.81 | 25.28 | 19.44 | 14.66 |
| LLaMA2-7B-Chat | 29.89 | 26.48 | 24.41 | 14.14 | 13.02 |
| LLaMA2-13B | 30.95 | 32.29 | 21.61 | 16.36 | 13.32 |
| LLaMA2-13B-Chat | 25.05 | 21.74 | 20.69 | 12.94 | 11.92 |
| ChatGLM2-6B | 28.45 | 25.07 | 20.27 | 19.86 | 19.71 |
| ChatGLM2-6B-32K | 19.25 | 18.86 | 20.35 | 15.16 | 13.04 |
| LongChat-7B-v1.5-32K | 27.57 | 28.78 | 26.30 | 18.98 | 23.14 |
| LongChat-13B-16K | 24.77 | 26.33 | 24.47 | 23.34 | 28.07 |
| Vicuna-7B-v1.5-16K | 32.52 | 31.99 | 26.03 | 21.18 | 20.79 |
| Vicuna-13B-v1.5-16K | 33.41 | 31.40 | 26.63 | 14.40 | 12.54 |
| GPT-3.5-Turbo-16K | 28.65 | 23.13 | 19.25 | 16.97 | 17.36 |

Table 16: BIGPATENT (SUM)

|  | 1k | 2k | 4k | 6k | 8k |
|---|---|---|---|---|---|
| LLaMA2-7B | 20.47 | 18.38 | 17.41 | 5.82 | 4.20 |
| LLaMA2-7B-Chat | 24.83 | 21.68 | 22.95 | 12.53 | 8.96 |
| LLaMA2-13B | 22.50 | 19.58 | 14.88 | 13.18 | 9.00 |
| LLaMA2-13B-Chat | 23.99 | 20.99 | 20.95 | 16.80 | 10.58 |
| ChatGLM2-6B | 23.07 | 20.42 | 16.81 | 16.39 | 15.74 |
| ChatGLM2-6B-32K | 22.13 | 19.25 | 18.57 | 17.72 | 17.53 |
| LongChat-7B-v1.5-32K | 25.92 | 23.51 | 20.52 | 14.96 | 17.83 |
| LongChat-13B-16K | 23.57 | 21.52 | 19.94 | 11.62 | 16.14 |
| Vicuna-7B-v1.5-16K | 27.63 | 23.65 | 23.53 | 19.24 | 16.77 |
| Vicuna-13B-v1.5-16K | 25.10 | 24.43 | 24.15 | 17.77 | 10.95 |
| GPT-3.5-Turbo-16K | 27.06 | 25.13 | 24.97 | 23.25 | 22.79 |

Table 17: Wikihow (SUM)

|  | 1k | 2k | 4k | 6k | 8k |
|---|---|---|---|---|---|
| LLaMA2-7B | 16.70 | 14.24 | 9.15 | 4.42 | 3.28 |
| LLaMA2-7B-Chat | 13.50 | 17.87 | 4.11 | 2.18 | 1.93 |
| LLaMA2-13B | 26.68 | 21.98 | 15.90 | 4.44 | 1.21 |
| LLaMA2-13B-Chat | 22.73 | 22.09 | 11.42 | 7.06 | 3.12 |
| ChatGLM2-6B | 16.90 | 15.23 | 13.05 | 13.65 | 12.20 |
| ChatGLM2-6B-32K | 20.92 | 21.94 | 18.73 | 16.93 | 15.77 |
| LongChat-7B-v1.5-32K | 19.33 | 25.59 | 18.80 | 11.03 | 7.14 |
| LongChat-13B-16K | 22.55 | 22.76 | 23.39 | 9.13 | 4.25 |
| Vicuna-7B-v1.5-16K | 15.87 | 21.25 | 8.34 | 10.64 | 5.55 |
| Vicuna-13B-v1.5-16K | 23.44 | 27.54 | 18.40 | 9.45 | 9.60 |
| GPT-3.5-Turbo-16K | 16.91 | 20.81 | 15.95 | 13.68 | 12.40 |

Table 18: Pubmed (SUM)

|  | 1k | 2k | 4k | 6k | 8k |
|---|---|---|---|---|---|
| LLaMA2-7B | 18.87 | 16.27 | 10.21 | 8.20 | 4.92 |
| LLaMA2-7B-Chat | 22.50 | 21.35 | 21.86 | 14.63 | 8.43 |
| LLaMA2-13B | 23.48 | 20.28 | 18.81 | 9.18 | 5.56 |
| LLaMA2-13B-Chat | 26.83 | 27.89 | 23.37 | 18.03 | 6.12 |
| ChatGLM2-6B | 24.96 | 20.87 | 9.54 | 2.28 | 0.53 |
| ChatGLM2-6B-32K | 23.39 | 22.91 | 24.64 | 22.35 | 19.76 |
| LongChat-7B-v1.5-32K | 24.47 | 24.58 | 24.07 | 19.53 | 13.33 |
| LongChat-13B-16K | 21.19 | 21.30 | 20.91 | 15.22 | 12.33 |
| Vicuna-7B-v1.5-16K | 24.71 | 25.92 | 24.31 | 17.50 | 18.67 |
| Vicuna-13B-v1.5-16K | 29.12 | 27.90 | 26.79 | 24.69 | 21.10 |
| GPT-3.5-Turbo-16K | 30.23 | 28.84 | 27.19 | 23.07 | 22.60 |

Table 19: NCLS (SUM)

|  | 1k | 2k | 4k | 6k | 8k |
|---|---|---|---|---|---|
| LLaMA2-7B | 10.00 | 17.00 | 16.50 | 10.00 | 9.50 |
| LLaMA2-7B-Chat | 8.00 | 11.00 | 17.00 | 14.00 | 12.00 |
| LLaMA2-13B | 7.00 | 15.50 | 16.50 | 12.63 | 11.00 |
| LLaMA2-13B-Chat | 17.50 | 24.00 | 18.50 | 13.42 | 11.00 |
| ChatGLM2-6B | 14.00 | 21.50 | 14.50 | 9.00 | 5.00 |
| ChatGLM2-6B-32K | 6.00 | 7.00 | 6.50 | 5.50 | 4.00 |
| LongChat-7B-v1.5-32K | 16.50 | 15.00 | 13.00 | 11.00 | 6.50 |
| LongChat-13B-16K | 23.50 | 22.50 | 21.50 | 23.50 | 12.00 |
| Vicuna-7B-v1.5-16K | 22.00 | 14.50 | 17.00 | 10.00 | 6.00 |
| Vicuna-13B-v1.5-16K | 13.00 | 16.00 | 16.50 | 11.00 | 13.04 |
| GPT-3.5-Turbo-16K | 19.50 | 19.50 | 20.00 | 18.50 | 14.50 |

Table 20: BIGPATENT (CLS)

|  | 1k | 2k | 4k | 6k | 8k |
|---|---|---|---|---|---|
| LLaMA2-7B | 7.78 | 0.01 | 0.00 | 0.00 | 0.03 |
| LLaMA2-7B-Chat | 4.03 | 0.28 | 0.01 | 0.00 | 0.00 |
| LLaMA2-13B | 13.19 | 0.90 | 2.89 | 0.19 | 0.00 |
| LLaMA2-13B-Chat | 7.48 | 1.19 | 0.01 | 0.00 | 0.00 |
| ChatGLM2-6B | 5.54 | 0.64 | 0.00 | 0.00 | 0.00 |
| ChatGLM2-6B-32K | 1.06 | 0.68 | 0.56 | 0.06 | 0.08 |
| LongChat-7B-v1.5-32K | 7.88 | 3.45 | 2.25 | 0.05 | 0.00 |
| LongChat-13B-16K | 5.60 | 1.82 | 0.59 | 0.00 | 0.00 |
| Vicuna-7B-v1.5-16K | 12.71 | 3.39 | 0.00 | 0.00 | 0.00 |
| Vicuna-13B-v1.5-16K | 15.56 | 11.69 | 6.55 | 0.02 | 0.00 |
| GPT-3.5-Turbo-16K | 21.60 | 20.01 | 19.40 | 16.32 | 11.17 |

Table 21: OpenSubtitles zh2en (TRAN)

|  | 1k | 2k | 4k | 6k | 8k |
|---|---|---|---|---|---|
| LLaMA2-7B | 6.14 | 0.95 | 0.00 | 0.00 | 0.00 |
| LLaMA2-7B-Chat | 8.30 | 3.04 | 0.73 | 0.20 | 0.00 |
| LLaMA2-13B | 9.17 | 3.68 | 1.40 | 0.21 | 0.01 |
| LLaMA2-13B-Chat | 12.77 | 8.00 | 0.97 | 0.00 | 0.00 |
| ChatGLM2-6B | 9.67 | 1.62 | 0.00 | 0.00 | 0.00 |
| ChatGLM2-6B-32K | 5.64 | 2.49 | 1.96 | 0.23 | 0.23 |
| LongChat-7B-v1.5-32K | 7.15 | 4.28 | 0.75 | 0.03 | 0.00 |
| LongChat-13B-16K | 4.69 | 2.61 | 2.06 | 0.58 | 0.00 |
| Vicuna-7B-v1.5-16K | 12.84 | 9.99 | 2.88 | 0.00 | 0.07 |
| Vicuna-13B-v1.5-16K | 15.60 | 13.52 | 10.05 | 2.23 | 1.38 |
| GPT-3.5-Turbo-16K | 20.61 | 21.18 | 23.13 | 21.28 | 19.57 |

Table 22: OpenSubtitles en2zh (TRAN)

|  | 1k | 2k | 4k | 6k | 8k |
|---|---|---|---|---|---|
| LLaMA2-7B | 9.50 | 4.98 | 3.50 | 2.46 | 0.48 |
| LLaMA2-7B-Chat | 14.50 | 4.98 | 0.50 | 0.00 | 0.00 |
| LLaMA2-13B | 11.50 | 8.96 | 1.00 | 0.99 | 0.00 |
| LLaMA2-13B-Chat | 15.50 | 3.48 | 0.00 | 0.99 | 0.00 |
| ChatGLM2-6B | 30.00 | 2.49 | 0.00 | 0.00 | 0.00 |
| ChatGLM2-6B-32K | 17.00 | 5.47 | 3.00 | 0.00 | 0.00 |
| LongChat-7B-v1.5-32K | 6.50 | 3.98 | 4.00 | 2.46 | 0.97 |
| LongChat-13B-16K | 13.50 | 4.98 | 6.00 | 5.91 | 0.00 |
| Vicuna-7B-v1.5-16K | 14.00 | 10.95 | 6.00 | 2.46 | 0.00 |
| Vicuna-13B-v1.5-16K | 40.00 | 23.88 | 7.00 | 0.00 | 0.00 |
| GPT-3.5-Turbo-16K | 38.00 | 22.89 | 11.50 | 5.91 | 5.31 |

Table 23: WikiText-103 (NLI)

|  | 1k | 2k | 4k | 6k | 8k |
|---|---|---|---|---|---|
| LLaMA2-7B | 24.00 | 22.00 | 1.49 | 0.50 | 0.49 |
| LLaMA2-7B-Chat | 39.00 | 30.00 | 0.50 | 0.50 | 0.00 |
| LLaMA2-13B | 47.50 | 22.50 | 0.50 | 4.00 | 0.00 |
| LLaMA2-13B-Chat | 66.00 | 5.00 | 0.00 | 0.00 | 0.00 |
| ChatGLM2-6B | 47.00 | 15.50 | 2.49 | 8.00 | 0.00 |
| ChatGLM2-6B-32K | 51.50 | 25.00 | 6.97 | 5.00 | 1.96 |
| LongChat-7B-v1.5-32K | 47.00 | 15.00 | 1.49 | 2.50 | 0.98 |
| LongChat-13B-16K | 21.50 | 23.50 | 1.00 | 5.00 | 0.00 |
| Vicuna-7B-v1.5-16K | 37.50 | 4.50 | 0.00 | 0.50 | 0.00 |
| Vicuna-13B-v1.5-16K | 75.00 | 26.00 | 3.48 | 0.00 | 0.00 |
| GPT-3.5-Turbo-16K | 77.50 | 58.00 | 4.98 | 12.50 | 4.41 |

Table 24: Wiki2019zh (NLI)

|  | 1k | 2k | 4k | 6k | 8k |
|---|---|---|---|---|---|
| LLaMA2-7B | 57.44 | 33.21 | 13.73 | 6.94 | 6.45 |
| LLaMA2-7B-Chat | 31.62 | 18.03 | 17.74 | 9.19 | 5.43 |
| LLaMA2-13B | 54.87 | 35.51 | 19.43 | 12.58 | 8.12 |
| LLaMA2-13B-Chat | 59.10 | 45.35 | 22.91 | 16.89 | 11.13 |
| ChatGLM2-6B | 45.35 | 34.06 | 9.15 | 8.68 | 0.87 |
| ChatGLM2-6B-32K | 20.92 | 10.02 | 17.49 | 15.33 | 12.09 |
| LongChat-7B-v1.5-32K | 48.47 | 43.76 | 32.78 | 25.66 | 21.02 |
| LongChat-13B-16K | 55.77 | 50.73 | 37.16 | 26.45 | 23.00 |
| Vicuna-7B-v1.5-16K | 52.23 | 43.40 | 30.19 | 18.55 | 10.60 |
| Vicuna-13B-v1.5-16K | 61.13 | 54.82 | 43.19 | 33.21 | 21.38 |
| GPT-3.5-Turbo-16K | 73.07 | 63.61 | 48.60 | 39.22 | 22.59 |

Table 25: MNDS News (CLS, Explicit Multiple)

|                      | 1k    | 2k    | 4k    | 6k    | 8k    |
|----------------------|-------|-------|-------|-------|-------|
| LLaMA2-7B            | 60.00 | 36.32 | 17.16 | 16.18 | 10.29 |
| LLaMA2-7B-Chat       | 30.00 | 27.86 | 22.55 | 19.12 | 12.00 |
| LLaMA2-13B           | 50.50 | 20.90 | 16.18 | 16.18 | 11.72 |
| LLaMA2-13B-Chat      | 43.50 | 43.78 | 26.96 | 28.89 | 19.57 |
| ChatGLM2-6B          | 47.50 | 34.33 | 17.65 | 15.20 | 15.69 |
| ChatGLM2-6B-32K      | 14.00 | 32.84 | 16.18 | 15.20 | 19.61 |
| LongChat-7B-v1.5-32K | 32.50 | 18.41 | 23.04 | 24.02 | 12.25 |
| LongChat-13B-16K     | 50.50 | 41.79 | 21.08 | 22.55 | 11.50 |
| Vicuna-7B-v1.5-16K   | 39.50 | 31.84 | 25.98 | 20.10 | 10.78 |
| Vicuna-13B-v1.5-16K  | 55.00 | 47.76 | 26.96 | 13.24 | 10.30 |
| GPT-3.5-Turbo-16K    | 54.50 | 39.80 | 17.65 | 19.61 | 12.25 |

Table 26: MNDS News (CLS, Semantic Multiple)

|                      | 1k    | 2k    | 4k    | 6k    | 8k    |
|----------------------|-------|-------|-------|-------|-------|
| LLaMA2-7B            | 29.00 | 19.92 | 12.00 | 6.35  | 2.19  |
| LLaMA2-7B-Chat       | 38.05 | 29.21 | 19.89 | 8.34  | 3.12  |
| LLaMA2-13B           | 47.18 | 43.22 | 16.05 | 2.65  | 0.00  |
| LLaMA2-13B-Chat      | 48.73 | 42.74 | 25.36 | 4.91  | 0.00  |
| ChatGLM2-6B          | 20.88 | 7.60  | 4.67  | 2.46  | 2.55  |
| ChatGLM2-6B-32K      | 20.54 | 8.85  | 6.01  | 0.22  | 0.00  |
| LongChat-7B-v1.5-32K | 34.88 | 30.98 | 26.39 | 6.88  | 0.00  |
| LongChat-13B-16K     | 51.43 | 44.99 | 30.75 | 7.94  | 0.00  |
| Vicuna-7B-v1.5-16K   | 33.63 | 29.48 | 6.49  | 0.23  | 0.00  |
| Vicuna-13B-v1.5-16K  | 66.40 | 45.69 | 32.44 | 21.28 | 11.65 |
| GPT-3.5-Turbo-16K    | 65.58 | 49.92 | 33.37 | 23.50 | 14.25 |

Table 27: MARC (CLS)

|                      | 1k    | 2k    | 4k    | 6k    | 8k    |
|----------------------|-------|-------|-------|-------|-------|
| LLaMA2-7B            | 31.60 | 21.02 | 22.52 | 17.92 | 15.95 |
| LLaMA2-7B-Chat       | 32.01 | 27.26 | 18.19 | 15.48 | 11.87 |
| LLaMA2-13B           | 40.79 | 33.70 | 27.80 | 16.87 | 12.38 |
| LLaMA2-13B-Chat      | 31.89 | 25.69 | 22.84 | 18.72 | 11.75 |
| ChatGLM2-6B          | 31.44 | 22.57 | 20.92 | 17.84 | 15.68 |
| ChatGLM2-6B-32K      | 37.68 | 30.31 | 29.33 | 22.77 | 17.71 |
| LongChat-7B-v1.5-32K | 30.79 | 28.92 | 23.22 | 15.25 | 9.19  |
| LongChat-13B-16K     | 26.88 | 24.92 | 23.17 | 14.93 | 12.08 |
| Vicuna-7B-v1.5-16K   | 32.74 | 29.45 | 25.10 | 16.76 | 11.08 |
| Vicuna-13B-v1.5-16K  | 35.06 | 32.61 | 31.64 | 23.05 | 19.39 |
| GPT-3.5-Turbo-16K    | 32.28 | 29.77 | 25.12 | 23.19 | 23.04 |

Table 28: DuReader (QA)

|                     | 1k    | 2k    | 4k    | 6k    | 8k    |
|---------------------|-------|-------|-------|-------|-------|
| LLaMA2-7B           | 23.80 | 6.10  | 0.72  | 0.09  | 0.05  |
| LLaMA2-7B-Chat      | 26.39 | 17.88 | 11.14 | 4.67  | 0.00  |
| LLaMA2-13B          | 43.50 | 22.64 | 10.20 | 2.85  | 0.00  |
| LLaMA2-13B-Chat     | 32.73 | 23.59 | 14.12 | 3.59  | 0.00  |
| ChatGLM2-6B         | 1.69  | 0.37  | 0.57  | 0.00  | 0.00  |
| ChatGLM2-6B-32K     | 10.22 | 3.87  | 0.89  | 0.00  | 0.00  |
| LongChat-7B-v1.5-32K| 28.13 | 19.17 | 10.14 | 4.72  | 0.00  |
| LongChat-13B-16K    | 27.78 | 16.21 | 3.11  | 1.28  | 0.00  |
| Vicuna-7B-v1.5-16K  | 19.58 | 6.93  | 0.20  | 0.10  | 0.43  |
| Vicuna-13B-v1.5-16K | 40.92 | 27.95 | 7.15  | 4.18  | 3.76  |
| GPT-3.5-Turbo-16K   | 34.84 | 31.15 | 19.03 | 14.29 | 10.23 |

Table 29: Online Shopping (CLS)

|                     | 1k    | 2k    | 4k    | 6k    | 8k    |
|---------------------|-------|-------|-------|-------|-------|
| LLaMA2-7B           | 67.17 | 33.62 | 20.27 | 7.54  | 4.00  |
| LLaMA2-7B-Chat      | 64.12 | 31.26 | 14.43 | 1.29  | 0.00  |
| LLaMA2-13B          | 58.83 | 34.57 | 16.17 | 4.71  | 0.00  |
| LLaMA2-13B-Chat     | 49.83 | 19.02 | 13.03 | 2.37  | 0.03  |
| ChatGLM2-6B         | 51.08 | 36.49 | 25.11 | 10.41 | 2.07  |
| ChatGLM2-6B-32K     | 67.03 | 40.79 | 16.10 | 10.50 | 5.99  |
| LongChat-7B-v1.5-32K| 39.75 | 22.85 | 9.40  | 2.97  | 0.00  |
| LongChat-13B-16K    | 44.00 | 15.12 | 6.97  | 1.10  | 2.96  |
| Vicuna-7B-v1.5-16K  | 45.75 | 21.52 | 5.87  | 1.33  | 0.00  |
| Vicuna-13B-v1.5-16K | 55.33 | 36.70 | 27.50 | 23.34 | 13.70 |
| GPT-3.5-Turbo-16K   | 75.75 | 77.28 | 59.08 | 47.32 | 44.98 |

Table 30: THUCNews (CLS, Explicit Multiple)

|                     | 1k    | 2k    | 4k    | 6k    | 8k    |
|---------------------|-------|-------|-------|-------|-------|
| LLaMA2-7B           | 54.00 | 50.00 | 21.50 | 21.08 | 16.67 |
| LLaMA2-7B-Chat      | 59.50 | 35.00 | 30.50 | 19.61 | 20.59 |
| LLaMA2-13B          | 63.50 | 38.50 | 24.50 | 20.76 | 19.52 |
| LLaMA2-13B-Chat     | 60.50 | 24.00 | 26.50 | 18.82 | 17.00 |
| ChatGLM2-6B         | 60.00 | 46.50 | 14.00 | 8.33  | 4.90  |
| ChatGLM2-6B-32K     | 61.00 | 38.00 | 23.00 | 13.24 | 11.69 |
| LongChat-7B-v1.5-32K| 38.50 | 29.50 | 30.00 | 13.73 | 11.38 |
| LongChat-13B-16K    | 46.50 | 38.50 | 22.00 | 16.37 | 16.67 |
| Vicuna-7B-v1.5-16K  | 58.50 | 30.00 | 17.00 | 12.39 | 10.73 |
| Vicuna-13B-v1.5-16K | 64.50 | 56.50 | 27.50 | 17.84 | 9.39  |
| GPT-3.5-Turbo-16K   | 61.00 | 44.50 | 18.50 | 14.71 | 11.27 |

Table 31: THUCNews (CLS, Semantic Multiple)

|  | 1k | 2k | 4k | 6k | 8k |
|---|---|---|---|---|---|
| LLaMA2-7B | 31.00 | 21.50 | 23.50 | 14.00 | 9.45 |
| LLaMA2-7B-Chat | 45.00 | 31.50 | 21.00 | 19.00 | 4.49 |
| LLaMA2-13B | 62.50 | 45.00 | 32.00 | 10.00 | 5.08 |
| LLaMA2-13B-Chat | 63.00 | 49.00 | 34.50 | 14.50 | 3.07 |
| ChatGLM2-6B | 38.00 | 26.50 | 16.00 | 7.50 | 0.50 |
| ChatGLM2-6B-32K | 55.50 | 52.00 | 42.50 | 33.50 | 19.85 |
| LongChat-7B-v1.5-32K | 24.00 | 27.50 | 21.00 | 19.00 | 10.31 |
| LongChat-13B-16K | 26.50 | 34.50 | 30.00 | 20.00 | 12.42 |
| Vicuna-7B-v1.5-16K | 37.00 | 36.00 | 32.50 | 19.00 | 11.39 |
| Vicuna-13B-v1.5-16K | 61.00 | 61.00 | 61.50 | 34.62 | 20.00 |
| GPT-3.5-Turbo-16K | 67.50 | 68.50 | 69.50 | 51.50 | 38.31 |

Table 32: THUCNews (CLS, Explicit Single)

|  | 1k | 2k | 4k | 6k | 8k |
|---|---|---|---|---|---|
| LLaMA2-7B | 18.50 | 14.00 | 5.00 | 4.48 | 3.50 |
| LLaMA2-7B-Chat | 30.50 | 22.50 | 10.50 | 11.44 | 3.50 |
| LLaMA2-13B | 33.50 | 35.50 | 8.50 | 7.46 | 2.00 |
| LLaMA2-13B-Chat | 35.50 | 36.50 | 15.00 | 10.95 | 8.50 |
| ChatGLM2-6B | 17.00 | 17.50 | 6.00 | 3.48 | 3.00 |
| ChatGLM2-6B-32K | 26.00 | 29.50 | 22.00 | 19.40 | 12.00 |
| LongChat-7B-v1.5-32K | 29.00 | 31.00 | 20.50 | 23.88 | 17.00 |
| LongChat-13B-16K | 32.00 | 34.00 | 31.00 | 15.47 | 8.00 |
| Vicuna-7B-v1.5-16K | 30.00 | 27.50 | 21.50 | 17.41 | 12.00 |
| Vicuna-13B-v1.5-16K | 40.50 | 38.50 | 34.50 | 20.40 | 12.00 |
| GPT-3.5-Turbo-16K | 41.50 | 41.50 | 33.00 | 26.37 | 17.50 |

Table 33: MNDS News (CLS, Explicit Single)

| 1k | 2k | 4k | 6k | 8k |
|---|---|---|---|---|
| LLaMA2-7B | 17.67 | 12.48 | 9.66 | 0.04 |
| LLaMA2-7B-Chat | 22.57 | 12.09 | 11.03 | 0.18 |
| LLaMA2-13B | 18.69 | 13.45 | 10.59 | 1.72 |
| LLaMA2-13B-Chat | 23.09 | 15.51 | 11.46 | 9.70 |
| ChatGLM2-6B | 28.61 | 14.23 | 10.56 | 9.45 |
| ChatGLM2-6B-32K | 28.13 | 18.41 | 11.73 | 15.73 |
| LongChat-7B-v1.5-32K | 21.11 | 14.99 | 11.63 | 7.21 |
| LongChat-13B-16K | 19.61 | 12.55 | 10.20 | 10.57 |
| Vicuna-7B-v1.5-16K | 17.09 | 14.54 | 12.07 | 20.21 |
| Vicuna-13B-v1.5-16K | 20.76 | 15.95 | 13.31 | 11.92 |
| GPT-3.5-Turbo-16K | 28.32 | 18.11 | 14.85 | 13.74 |

Table 34: CNewsum (SUM)

| | 1k | 2k | 4k | 6k | 8k |
|---|---|---|---|---|---|
| LLaMA2-7B | 37.12 | 26.96 | 24.15 | 10.31 | 8.68 |
| LLaMA2-7B-Chat | 36.83 | 31.13 | 12.40 | 11.31 | 7.94 |
| LLaMA2-13B | 33.86 | 28.09 | 20.15 | 12.96 | 9.20 |
| LLaMA2-13B-Chat | 34.12 | 26.76 | 23.76 | 17.05 | 10.34 |
| ChatGLM2-6B | 37.26 | 23.70 | 10.97 | 8.89 | 10.06 |
| ChatGLM2-6B-32K | 38.11 | 34.49 | 32.31 | 29.36 | 26.12 |
| LongChat-7B-v1.5-32K | 39.25 | 32.58 | 26.72 | 23.24 | 19.26 |
| LongChat-13B-16K | 37.34 | 32.63 | 26.10 | 23.62 | 19.00 |
| Vicuna-7B-v1.5-16K | 34.73 | 30.68 | 27.81 | 17.40 | 20.11 |
| Vicuna-13B-v1.5-16K | 34.16 | 30.03 | 27.68 | 10.56 | 9.88 |
| GPT-3.5-Turbo-16K | 37.81 | 32.25 | 30.26 | 26.23 | 25.09 |

Table 35: CLTS+ (SUM)

| | 1k | 2k | 4k | 6k | 8k |
|---|---|---|---|---|---|
| LLaMA2-7B | 20.99 | 20.96 | 16.51 | 9.00 | 8.88 |
| LLaMA2-7B-Chat | 20.58 | 19.72 | 16.87 | 10.08 | 7.75 |
| LLaMA2-13B | 21.30 | 20.92 | 14.27 | 7.71 | 4.00 |
| LLaMA2-13B-Chat | 21.22 | 19.83 | 17.50 | 8.50 | 3.83 |
| ChatGLM2-6B | 25.08 | 24.62 | 20.53 | 17.22 | 14.85 |
| ChatGLM2-6B-32K | 22.77 | 23.36 | 22.19 | 21.99 | 18.64 |
| LongChat-7B-v1.5-32K | 21.28 | 21.16 | 21.08 | 15.63 | 6.56 |
| LongChat-13B-16K | 20.48 | 21.11 | 20.57 | 12.52 | 8.00 |
| Vicuna-7B-v1.5-16K | 22.21 | 21.05 | 19.97 | 15.67 | 4.99 |
| Vicuna-13B-v1.5-16K | 21.70 | 21.72 | 21.98 | 21.65 | 11.29 |
| GPT-3.5-Turbo-16K | 25.08 | 24.56 | 24.52 | 22.51 | 22.19 |

Table 36: CEPSUM (SUM)

| | 1k | 2k | 4k | 6k | 8k |
|---|---|---|---|---|---|
| LLaMA2-7B | 18.75 | 15.32 | 13.38 | 11.23 | 9.84 |
| LLaMA2-7B-Chat | 16.69 | 9.00 | 3.98 | 2.12 | 3.23 |
| LLaMA2-13B | 17.71 | 15.68 | 7.67 | 5.06 | 5.31 |
| LLaMA2-13B-Chat | 9.90 | 9.37 | 5.14 | 4.48 | 3.12 |
| ChatGLM2-6B | 10.84 | 18.96 | 14.35 | 14.14 | 10.39 |
| ChatGLM2-6B-32K | 18.86 | 18.26 | 19.39 | 18.49 | 11.71 |
| LongChat-7B-v1.5-32K | 12.74 | 15.36 | 17.57 | 29.64 | 3.59 |
| LongChat-13B-16K | 10.41 | 11.74 | 16.29 | 12.32 | 4.85 |
| Vicuna-7B-v1.5-16K | 14.15 | 19.49 | 21.00 | 12.65 | 5.52 |
| Vicuna-13B-v1.5-16K | 18.46 | 21.13 | 19.08 | 17.37 | 15.32 |
| GPT-3.5-Turbo-16K | 13.39 | 12.35 | 11.70 | 14.23 | 11.27 |

Table 37: CNNNews (SUM)

|  | 1k | 2k | 4k | 6k | 8k |
|---|---|---|---|---|---|
| LLaMA2-7B | 5.00 | 10.15 | 11.64 | 7.03 | 4.20 |
| LLaMA2-7B-Chat | 10.04 | 7.44 | 3.49 | 2.23 | 2.88 |
| LLaMA2-13B | 11.75 | 9.14 | 10.58 | 8.88 | 5.12 |
| LLaMA2-13B-Chat | 9.84 | 6.27 | 8.39 | 8.34 | 7.06 |
| ChatGLM2-6B | 13.91 | 13.99 | 15.63 | 12.42 | 12.93 |
| ChatGLM2-6B-32K | 13.21 | 15.08 | 12.26 | 12.10 | 11.86 |
| LongChat-7B-v1.5-32K | 10.14 | 9.95 | 9.84 | 6.96 | 2.08 |
| LongChat-13B-16K | 8.78 | 9.17 | 13.77 | 7.53 | 1.21 |
| Vicuna-7B-v1.5-16K | 10.89 | 11.51 | 12.07 | 8.88 | 2.16 |
| Vicuna-13B-v1.5-16K | 9.75 | 13.49 | 20.83 | 12.42 | 10.70 |
| GPT-3.5-Turbo-16K | 16.72 | 15.51 | 15.88 | 15.35 | 16.45 |

Table 38: News2016 (SUM)

|  | 1k | 2k | 4k | 6k | 8k |
|---|---|---|---|---|---|
| LLaMA2-7B | 22.34 | 18.79 | 9.45 | 8.31 | 4.36 |
| LLaMA2-7B-Chat | 19.78 | 20.01 | 11.21 | 9.41 | 5.39 |
| LLaMA2-13B | 20.62 | 16.49 | 5.05 | 3.26 | 4.31 |
| LLaMA2-13B-Chat | 22.19 | 19.63 | 11.53 | 8.41 | 7.12 |
| ChatGLM2-6B | 23.78 | 26.44 | 17.99 | 11.52 | 8.16 |
| ChatGLM2-6B-32K | 21.24 | 14.47 | 16.09 | 11.31 | 9.71 |
| LongChat-7B-v1.5-32K | 21.34 | 19.03 | 19.89 | 11.73 | 5.62 |
| LongChat-13B-16K | 19.41 | 17.68 | 15.97 | 11.25 | 8.13 |
| Vicuna-7B-v1.5-16K | 21.70 | 20.32 | 22.22 | 7.13 | 7.13 |
| Vicuna-13B-v1.5-16K | 21.93 | 21.84 | 20.60 | 11.17 | 8.52 |
| GPT-3.5-Turbo-16K | 27.46 | 27.34 | 21.02 | 12.98 | 11.97 |

Table 39: LCSTS (SUM)

|  | 1k | 2k | 4k | 6k | 8k |
|---|---|---|---|---|---|
| LLaMA2-7B | 31.50 | 28.00 | 23.88 | 16.00 | 5.45 |
| LLaMA2-7B-Chat | 35.50 | 30.50 | 19.90 | 14.50 | 8.98 |
| LLaMA2-13B | 37.50 | 34.00 | 29.85 | 7.50 | 5.00 |
| LLaMA2-13B-Chat | 44.50 | 42.50 | 34.33 | 16.83 | 13.21 |
| ChatGLM2-6B | 71.00 | 66.50 | 61.19 | 24.21 | 8.95 |
| ChatGLM2-6B-32K | 72.50 | 70.50 | 63.18 | 58.00 | 43.49 |
| LongChat-7B-v1.5-32K | 30.00 | 30.00 | 25.87 | 26.50 | 10.26 |
| LongChat-13B-16K | 23.00 | 29.00 | 24.38 | 30.00 | 18.96 |
| Vicuna-7B-v1.5-16K | 34.50 | 27.50 | 26.87 | 21.00 | 12.82 |
| Vicuna-13B-v1.5-16K | 56.00 | 49.50 | 52.74 | 50.00 | 30.98 |
| GPT-3.5-Turbo-16K | 85.00 | 84.00 | 81.09 | 76.00 | 74.02 |

Table 40: C3 (QA)

|  | 1k | 2k | 4k | 6k | 8k |
|---|---|---|---|---|---|
| LLaMA2-7B | 2.50 | 1.00 | 0.00 | 1.99 | 4.50 |
| LLaMA2-7B-Chat | 7.50 | 2.00 | 0.50 | 2.49 | 7.50 |
| LLaMA2-13B | 6.00 | 3.50 | 2.00 | 1.00 | 0.00 |
| LLaMA2-13B-Chat | 7.50 | 7.50 | 4.50 | 3.30 | 0.00 |
| ChatGLM2-6B | 8.50 | 7.00 | 8.00 | 5.47 | 4.00 |
| ChatGLM2-6B-32K | 9.50 | 8.00 | 9.00 | 6.97 | 8.00 |
| LongChat-7B-v1.5-32K | 13.50 | 16.00 | 15.50 | 14.93 | 4.84 |
| LongChat-13B-16K | 8.50 | 7.50 | 16.00 | 11.44 | 7.50 |
| Vicuna-7B-v1.5-16K | 7.50 | 11.00 | 8.00 | 6.97 | 1.61 |
| Vicuna-13B-v1.5-16K | 11.00 | 19.00 | 24.50 | 14.93 | 4.00 |
| GPT-3.5-Turbo-16K | 18.00 | 16.00 | 13.00 | 14.43 | 18.50 |

Table 41: NewsQA (QA)

|  | 1k | 2k | 4k | 6k | 8k |
|---|---|---|---|---|---|
| LLaMA2-7B | 38.00 | 31.00 | 26.50 | 19.00 | 10.50 |
| LLaMA2-7B-Chat | 41.00 | 37.00 | 34.00 | 24.00 | 10.00 |
| LLaMA2-13B | 41.00 | 36.00 | 29.00 | 24.00 | 12.50 |
| LLaMA2-13B-Chat | 42.50 | 42.50 | 34.50 | 30.50 | 18.00 |
| ChatGLM2-6B | 35.50 | 27.00 | 12.00 | 12.00 | 8.00 |
| ChatGLM2-6B-32K | 31.50 | 32.50 | 29.50 | 27.00 | 18.50 |
| LongChat-7B-v1.5-32K | 43.50 | 42.50 | 37.50 | 33.00 | 16.50 |
| LongChat-13B-16K | 43.00 | 37.00 | 35.50 | 32.00 | 17.50 |
| Vicuna-7B-v1.5-16K | 42.00 | 40.50 | 35.00 | 31.00 | 20.00 |
| Vicuna-13B-v1.5-16K | 39.50 | 38.00 | 36.00 | 28.00 | 21.50 |
| GPT-3.5-Turbo-16K | 39.50 | 36.50 | 32.50 | 31.00 | 32.50 |

Table 42: Duorc (QA)

|  | 1k | 2k | 4k | 6k | 8k |
|---|---|---|---|---|---|
| LLaMA2-7B | 10.27 | 6.66 | 2.20 | 2.01 | 0.69 |
| LLaMA2-7B-Chat | 8.83 | 5.13 | 1.37 | 1.13 | 0.40 |
| LLaMA2-13B | 20.99 | 12.85 | 2.92 | 1.78 | 0.72 |
| LLaMA2-13B-Chat | 15.93 | 9.24 | 3.64 | 2.58 | 1.32 |
| ChatGLM2-6B | 12.85 | 7.61 | 0.28 | 0.69 | 0.38 |
| ChatGLM2-6B-32K | 13.44 | 5.05 | 3.60 | 3.37 | 3.22 |
| LongChat-7B-v1.5-32K | 14.10 | 10.97 | 8.00 | 6.39 | 4.78 |
| LongChat-13B-16K | 10.40 | 8.85 | 5.13 | 4.54 | 3.24 |
| Vicuna-7B-v1.5-16K | 19.88 | 20.31 | 8.61 | 7.74 | 3.17 |
| Vicuna-13B-v1.5-16K | 27.31 | 22.04 | 13.88 | 9.82 | 5.13 |
| GPT-3.5-Turbo-16K | 33.30 | 28.38 | 24.33 | 23.94 | 18.48 |

Table 43: News Commentary en2zh (TRAN)

|  | 1k | 2k | 4k | 6k | 8k |
|---|---|---|---|---|---|
| LLaMA2-7B | 13.28 | 7.42 | 0.89 | 0.22 | 0.01 |
| LLaMA2-7B-Chat | 8.16 | 4.01 | 0.50 | 0.32 | 0.09 |
| LLaMA2-13B | 20.28 | 13.89 | 5.43 | 3.32 | 1.48 |
| LLaMA2-13B-Chat | 28.83 | 17.19 | 7.53 | 5.56 | 3.19 |
| ChatGLM2-6B | 6.80 | 7.51 | 0.16 | 0.04 | 0.02 |
| ChatGLM2-6B-32K | 5.55 | 7.32 | 1.14 | 2.26 | 2.21 |
| LongChat-7B-v1.5-32K | 15.01 | 9.61 | 7.31 | 2.91 | 3.08 |
| LongChat-13B-16K | 12.82 | 9.55 | 4.18 | 2.30 | 1.23 |
| Vicuna-7B-v1.5-16K | 17.64 | 15.14 | 10.58 | 6.76 | 2.35 |
| Vicuna-13B-v1.5-16K | 20.17 | 17.43 | 12.88 | 7.82 | 4.35 |
| GPT-3.5-Turbo-16K | 26.23 | 22.22 | 17.99 | 15.94 | 13.12 |

Table 44: News Commentary zh2en (TRAN)

|  | 1k | 2k | 4k | 6k | 8k |
|---|---|---|---|---|---|
| LLaMA2-7B | 9.30 | 6.21 | 1.01 | 0.91 | 1.05 |
| LLaMA2-7B-Chat | 15.20 | 9.40 | 3.05 | 2.17 | 0.88 |
| LLaMA2-13B | 14.58 | 10.47 | 2.71 | 3.00 | 1.21 |
| LLaMA2-13B-Chat | 13.94 | 10.78 | 2.16 | 3.09 | 1.49 |
| ChatGLM2-6B | 14.86 | 0.98 | 0.07 | 0.02 | 0.00 |
| ChatGLM2-6B-32K | 13.67 | 5.19 | 1.84 | 1.17 | 1.18 |
| LongChat-7B-v1.5-32K | 20.43 | 9.78 | 4.23 | 2.93 | 3.03 |
| LongChat-13B-16K | 6.43 | 5.50 | 2.91 | 2.06 | 1.36 |
| Vicuna-7B-v1.5-16K | 23.75 | 11.36 | 5.93 | 2.01 | 3.23 |
| Vicuna-13B-v1.5-16K | 22.52 | 20.22 | 9.77 | 4.03 | 2.53 |
| GPT-3.5-Turbo-16K | 25.84 | 22.48 | 13.99 | 9.84 | 9.39 |

Table 45: Tedtalks en2zh (TRAN)

|  | 1k | 2k | 4k | 6k | 8k |
|---|---|---|---|---|---|
| LLaMA2-7B | 13.82 | 5.32 | 0.25 | 0.00 | 0.00 |
| LLaMA2-7B-Chat | 17.49 | 5.26 | 1.99 | 0.93 | 0.00 |
| LLaMA2-13B | 19.94 | 5.55 | 1.75 | 0.00 | 8.00 |
| LLaMA2-13B-Chat | 17.37 | 5.74 | 2.64 | 0.00 | 0.00 |
| ChatGLM2-6B | 13.22 | 4.26 | 1.03 | 0.19 | 0.05 |
| ChatGLM2-6B-32K | 9.72 | 2.91 | 1.53 | 1.77 | 1.31 |
| LongChat-7B-v1.5-32K | 12.06 | 2.01 | 0.43 | 0.09 | 0.00 |
| LongChat-13B-16K | 14.78 | 2.05 | 0.99 | 1.11 | 0.48 |
| Vicuna-7B-v1.5-16K | 20.46 | 5.97 | 1.97 | 0.83 | 0.00 |
| Vicuna-13B-v1.5-16K | 24.07 | 11.94 | 7.27 | 5.79 | 3.48 |
| GPT-3.5-Turbo-16K | 16.14 | 10.86 | 9.32 | 7.85 | 4.46 |

Table 46: Tedtalks zh2en (TRAN)

