# OpenReview forum: "M$^4$LE: A Multi-Ability Multi-Range Long Context Evaluation Benchmark for Large Language Models"
_ICLR.cc/2024/Conference — Submitted to ICLR 2024_

### Official Review · Reviewer_qZry · 2023-11-01

**Soundness:** 3 good
**Presentation:** 3 good
**Contribution:** 3 good
**Rating:** 5
**Confidence:** 5

**Summary:**

In this paper, the authors propose M4LE, which is a multi-ability, multi-range, multi-task, multi-domain long-context evaluation benchmark for large language models. It evaluates the ability of LLMs to understand long sequences and perform tasks in different languages and context lengths. The benchmark includes five different abilities and five different context length ranges. The results show that current LLMs still struggle to understand long-context information, and the performance of different models varies in different context length ranges and languages. The benchmark also delves into the factors influencing long-context understanding capability, including the positioning of relevant information and the performance of LLMs under different languages.

**Strengths:**

1. The main contribution of the paper is a new long-context benchmark which covers a wide range of tasks. M4LE can better evaluate the long-context ability of LLMs across various scenarios.

2. The author evaluates 11 widely well-known LLMs on the M4LE benchmark, and find some interesting conclusions.

3. The authors do experiments with their benchmark to verify the lost-in-the-middle phenomenon in LLMs.

**Weaknesses:**

It is better to include some comparison between M4LE and other benchmarks in Table 2. It is also good to include the max. length number in it.

The observations from the evaluation in the paper are weak and uninformative, except that the existing LLMs struggle to process long inputs, and ChatGPT performs the strongest among all. These can be shown from other existing benchmarks as well.

The categorization and organization of different ranges of sentences lengths can also be done in other similar benchmarks.

The most interesting point introduced in the paper is to categorize inputs to 5 abilities, explicit single-span, semantic multi-span understanding. However, it is unfortunate that there is an unclear analysis with little informative points in the paper in terms of this point.

List a few unclear points.
1.	For example, M4LE spans different domains and tasks, while the evaluation does not consider these points, e.g. specific analysis on the different impacts of QA, translation.

2.	Besides, it shows the situations from 3 types of tasks are different. There can be more sophisticated explanations and experiments to furnish further ideas.

3.	In Figure 2, why most models perform worse on explicit multi-span understanding compared to both semantic multi-span and global context understanding?

4.	From the results in the paper, ChatGPT is the strongest ones in processing long-context inputs, which is much large in its size. However, only 13B models are evaluated in the paper. It is still unclear that whether larger models can perform much better, e.g. LLaMA2-70b.

5.. The paper seems to have no detailed report about how to construct the benchmark, for example, which instances I will consider to combine them together, or I will choose them randomly? how the number of instances N is decided? In that case, I will think that
the M4LE just simply combine the small datasets together to build long-context datasets, where most of them have no relevance between each other. It means that M4LE is a benchmark mainly tests the retrieval ability over long-context but less consider the comprehensive understanding over the global context.

6. In the experiments on different language, the authors claim that Vicuna and Long-chat exhibit a more pronounced performance drop in Chinese. However, in Figure 4, the drops of these models between these two languages have no obvious difference. For example, Vicuna-13B-1.5-16k declines greatly both in English and Chinese after 4k length.


Maybe I miss something. I am confused that M4LE is seemingly built from existing web sources, which LLMs are exposed to during training. How does it alleviate the data leakage problem, which also exists for other benchmarks?

**Questions:**

see my weakness part

---

> ### Author Response · Authors · 2023-11-16
>
> Thank you for your insightful comments and suggestions. We have carefully considered your feedback and have addressed each point as follows:
>
> **Q1: It is better to include some comparison between M4LE and other benchmarks in Table 2. It is also good to include the max. length number in it.**
>
> We appreciate your suggestion and have accordingly included detailed statistics in the revised paper under Appendix A.1.
>
> **Q2: The evaluation in the paper are weak and uninformative, except that the existing LLMs struggle to process long inputs, and ChatGPT performs the strongest among all. These can be shown from other existing benchmarks as well.
> The categorization and organization of different ranges of sentence lengths can also be done in other similar benchmarks.**
>
> Our paper introduces the first comprehensive benchmark for long-context evaluation, systematically designed around five key abilities that encompass a broad spectrum of real-world scenarios, with contexts extending up to 32K words. Therefore, it is important to examine the performance of different LLMs on this comprehensive benchmark. Additionally, our analysis delves into performance across different languages and investigates the 'lost-in-the-middle' phenomenon, as reported by Liu et al. (2023). We also present further analysis of performance across various task types, as mentioned below.
>
> **Q3: M4LE spans different domains and tasks, while the evaluation does not consider these points**
>
> Thank you for your valuable feedback. In response, we have added a new section, "Appendix A.6: Performance in Various Task Types", to our revised paper. We also want to highlight that our benchmark, designed to encompass a broad spectrum of real-world scenarios with long inputs, is based on five key long-context abilities. This benchmark, which considers various domains and tasks, ensures comprehensive coverage of different use cases. Therefore, our main paper focuses primarily on analyses related to these five abilities.
>
> **Q4: it shows the situations from 3 types of tasks are different. There can be more sophisticated explanations and experiments to furnish further ideas.**
>
>
> We seek clarification on whether you are referring to Figure 3, which shows the effect of placing the supporting document at different locations. While we acknowledge the subtle differences highlighted in Figure 3, our paper's primary focus is on a methodology for assessing the performance variations with context length. Therefore, we think it would be best for future work to perform more extensive experiments (More task types, more datasets per task type, different setup, etc.).  As for the performance variations in different tasks of our main experiments, we have included them in the revised paper as stated in the previous question.
>
> **Q5: why most models perform worse on explicit multi-span understanding compared to both semantic multi-span and global context understanding**
>
> Thank you for raising this interesting question!  We briefly discussed this phenomenon in Section 4.3 under "Multiple-span understanding is more difficult, and semantic retrieval is even harder for competent models." While we think that more work is needed to investigate this, we hypothesize that explicit multi-span understanding requires the model to generate hexadecimal identifiers is an extremely unfamiliar task for LLMs. It is very different from most pre-training tasks, and therefore only the most capable LLMs that were fine-tuned on massively diverse tasks and aligned well with human preference can generalize and demonstrate adequate performance at this task.
>
>
> **Q6: only 13B models are evaluated in the paper. It is still unclear whether larger models can perform much better, e.g. LLaMA2-70b.**
>
> We are in the process of extending our experiments to include larger models like LLaMA2-70b. Due to the extensive computational resources required for these experiments, we anticipate some delay in obtaining and analyzing the results. We are committed to including this data in our study as soon as it is available and will provide updates accordingly.
>
> **Q7: The paper seems to have no detailed report about how to construct the benchmark**
>
> Thank you for your suggestions. We have included more details in "Section 3.3 Task Construction". Specifically, we determine the number of instances by dividing the desired context length with the median length of the instances in the testing set. Then,  we sample these instances randomly from the testing set.

---

> ### Author Response · Authors · 2023-11-16
>
> **Q8: I will think that the M4LE just simply combine the small datasets together to build long-context datasets, where most of them have no relevance between each other. It means that M4LE is a benchmark mainly tests the retrieval ability over long-context but less consider the comprehensive understanding over the global context.**
>
> Thank you for your comment regarding the M4LE. I would like to take this opportunity to clarify the design of M4LE. M4LE does not just test the retrieval ability over long-context.  It contains 22 tasks that target the semantic and global understanding ability of the models.
> These tasks require LLM to understand the full context thoroughly in order to retrieve relevant information to complete the task.
> For tasks targeting semantic understanding, the model needs to identify a few key text spans after it comprehends the whole context to finish the task. Tasks focused on global understanding require the model to integrate and interpret information from the full context to complete the task.
> Therefore, by targeting the five different abilities, M4LE is a comprehensive benchmark that evaluates a wide range of long-context understanding scenarios.
>
> **Q9: The authors claim that Vicuna and Long-chat exhibit a more pronounced performance drop in Chinese. However, in Figure 4, the drops of these models between these two languages have no obvious difference.**
>
> Thank you for your observation and comment! We originally computed the slope of the best-fit lines for both languages and found that Vicuna and LongChat models indeed have steeper slope in Chinese. However, after the updates on the results as mentioned in the global response, we observed similar slopes for both languages. Therefore, we updated the paper to reflect that.
>
> **Q10 : How does it alleviate the data leakage problem, which also exists for other benchmarks?**
>
> Thank you for your question. Indeed, it is an important problem for benchmarking, as also mentioned by Reviewer "sezj".
>
> Our methodology combines the shorter texts which actually present an instance that is sufficiently distinct to the original ones. For instance, in the WikiText-103 (NLI) task, we observed that even though Wikipedia articles are commonly included in language model training datasets, the performance of most models is poor even when the input contexts are limited to 1k tokens. This suggests that the benchmark effectively challenges the models beyond their training conditions.
>
> Besides, changing the question type and format, for example by asking the models to return the hexadecimal identifier of all the articles that belong to a certain class presents a different challenge for different LLMs, as evidenced by the poor performance of most models in MNDS News (Explicit Multiple) and THUCNews (Explicit Multiple).
>
> We hope these revisions and clarifications address your concerns adequately. Thank you for your insightful feedback, which has significantly contributed to the improvement of our paper.

---

> ### Author Response · Authors · 2023-11-23
>
> Thank you for your valuable time and feedback. As the deadline for author-reviewer discussion ends today, we kindly request that you acknowledge our rebuttal and engage in further discussion if you have any additional comments.

---

### Official Review · Reviewer_NJ6x · 2023-11-01

**Soundness:** 3 good
**Presentation:** 3 good
**Contribution:** 3 good
**Rating:** 6
**Confidence:** 4

**Summary:**

This paper proposes a new Multi-ability, Multirange, Multi-task, Multi-domain benchmark called M^4LE for the long-sequence ability assessment of LLMs. The experiments on multiple popular LLMs reveal shortcomings in their ability to handle long-text inputs from various perspectives.

**Strengths:**

1. The collected data in this paper is extensive, encompassing 36 datasets from 12 tasks and domains, enabling the comprehensive evaluation of long-text generation by existing models across 5 different abilities. The considered task types and model capabilities are also highly diverse.

2. The evaluation experiments on several existing LLMs for both Chinese and English are substantial. They reveal from multiple angles the existing models' shortcomings in handling long-text inputs.

**Weaknesses:**

Additional statistical information of the proposed dataset could be provided, such as the distribution of sentence lengths and the proportion of samples in different languages.

**Questions:**

Do the authors have a data cleaning process? How is it done?

---

> ### Author Response · Authors · 2023-11-16
>
> Thank you for your insightful comments and suggestions. We have carefully considered your feedback and have addressed each point as follows:
>
> **Q1: Additional statistical information of the proposed dataset could be provided, such as the distribution of sentence lengths and the proportion of samples in different languages.**
>
> We have now included detailed statistics in the revised paper under Appendix A.1. We believe these additional details will provide a clearer understanding of the dataset's composition and diversity.
>
>
> **Q2: Do the authors have a data cleaning process? How is it done?**
> Regarding our data cleaning process, we adhered to the preprocessing process of the original datasets. Additionally, we exclude instances that were either excessively long or short for our benchmark. The precise methodology and criteria used for this filtering process have been outlined in detail in Appendix A.1 of our original submission.
>
> We appreciate your constructive feedback, which has significantly contributed to enhancing the quality of our paper. We hope that our revisions meet your expectations and look forward to any further suggestions you may have.

---

> ### Author Response · Authors · 2023-11-23
>
> Thank you for your valuable time and feedback. As the deadline for author-reviewer discussion ends today, we kindly request that you acknowledge our rebuttal and engage in further discussion if you have any additional comments.

---

### Official Review · Reviewer_2qJD · 2023-11-01

**Soundness:** 1 poor
**Presentation:** 3 good
**Contribution:** 2 fair
**Rating:** 5
**Confidence:** 4

**Summary:**

The paper introduces M4LE, a comprehensive benchmark for evaluating the long-sequence capability of large language models (LLMs). M4LE comprises a diverse set of 36 NLP datasets, covering 12 task types and 12 domains, to address the scarcity of tasks with naturally long sequences. The benchmark incorporates five different types of abilities, including explicit and semantic single-span, explicit and semantic multiple-span, and global context understanding. This work conducts massive experiments to evaluate existing large language models with the proposed M4LE. The study also explores the impact of factors such as language differences and the positioning of relevant information on long-context understanding capabilities.

**Strengths:**

1. M4LE offers a multi-dimension assessment by including tasks that cover different abilities, ranges, tasks and domains. This makes it feasible to evaluate a wide variety of skills for long context LMs.

2. M4LE explores factors such as language differences and the positioning of relevant information, showing their impact on the models' long-context understanding capabilities. This analysis contributes to a deeper understanding of the challenges and potential improvements in handling long context inputs.

3. The paper was, in general, easy to follow. Its motivation is reasonable (but see the weakness).

**Weaknesses:**

1. My primary concern pertains to the strategy of constructing long sequences by simply converting short sequences into longer ones. It is clear that the model's performance deteriorates significantly as the length of the context increases, which raises doubts about the effectiveness and appropriateness of such data formation. The random selection of original short sequences appears to negatively impact the performance. Further investigation is needed to determine how to control the variance of data distributions/domains within a long sequence of text and assess the validity and reasonability of this data formation approach.

2. Additionally, I kindly request a performance comparison with recent state-of-the-art models, such as Llama2-4k.

**Questions:**

1. Please address the weaknesses above.

---

> ### Author Response · Authors · 2023-11-16
>
> Thank you for your insightful comments and thorough evaluation of our paper. We are grateful for the opportunity to address your concerns and provide additional clarity regarding our study. Please find our detailed responses below:
>
> **Q1: It is clear that the model's performance deteriorates significantly as the length of the context increases, which raises doubts about the effectiveness and appropriateness of such data formation. The random selection of original short sequences appears to negatively impact the performance.**
>
> You raised a critical point regarding the construction of long sequences from shorter ones, and the potential impact of this methodology on model performance. We agree that the performance degradation observed in models as context length increases highlights the challenges LLMs face in processing long sequences. However, we argue that this actually supports that our approach is effective rather than a flaw in our data formation approach.
>
> Our method of extending short sequences into longer contexts is designed to rigorously test the five long-context understanding abilities of LLMs without building new datasets from scratch. The performance deterioration observed in some models, especially at longer context lengths, is indicative of their current limitations in handling extended sequences. This is an important aspect of our benchmark, as it successfully distinguishes between models based on their ability to process long contexts. For instance, as illustrated in Figure 2, more competent models like GPT-3.5-Turbo-16K and ChatGLM2-6B-32K exhibit a relatively smaller drop in performance, demonstrating their stronger long-context capabilities.
>
> Additionally, we acknowledge the concern about the random selection of original short sequences. We want to clarify that our approach also aims to mitigate potential pre-training biases inherent in LLMs. Therefore, the difficulty models face with instances formed with randomly selected sequences underscores the effectiveness of our approach in providing a more unbiased assessment of their long-context understanding abilities.
>
> **Q2: I kindly request a performance comparison with recent state-of-the-art models, such as Llama2-4k.**
>
> Regarding your request for a performance comparison with the latest state-of-the-art models, such as Llama2-4k, we appreciate the importance of including such comparisons in our study. In our initial submission, we evaluated the Llama2-4k models, specifically the 7B and 13B variants, alongside their chat versions.
>
> For Llama2-70B models, we are currently conducting experiments with these models. However, due to the extensive computational resources required for these experiments, we anticipate some delay in obtaining and analyzing the results. We are committed to including this data in our study as soon as it is available and will provide updates accordingly.
>
> We hope our responses adequately address your concerns. We are committed to advancing the understanding of LLMs in processing long sequences and believe our benchmark, M4LE, contributes significantly to this field. We look forward to any further feedback you may have.

---

> > ### Comment · Reviewer_2qJD · 2023-11-22
> >
> > Thank you for the response. I have no further questions and keep the rating unchanged.

---

### Official Review · Reviewer_sezj · 2023-11-01

**Soundness:** 3 good
**Presentation:** 3 good
**Contribution:** 3 good
**Rating:** 3
**Confidence:** 4

**Summary:**

The paper introduces M4LE, an evaluation benchmark designed to assess the long-sequence understanding capabilities of large language models (LLMs). Recognising the limitation of current benchmarks, which primarily focus on short sequences, the authors propose a multi-ability, multi-range, multi-task, multi-domain evaluation strategy. M4LE is based on a diverse NLP task pool consisting of 36 NLP datasets, 12 task types, and 12 domains. The authors propose an automatic approach to convert short-sequence tasks into long-sequence scenarios.

**Strengths:**

M4LE provides a comprehensive evaluation of LLMs' long-context understanding capabilities across different abilities and length ranges. This is a significant advancement over existing benchmarks which primarily target short sequences.

The authors have collected a diverse set of tasks from a variety of domains which makes the benchmark more robust and comprehensive.

**Weaknesses:**

1. The construction of the dataset is detached from realistic scenarios. This is because the dataset is mainly composed of short texts synthesized into longer documents, which is not typically how users interact with long-context language models. They are unlikely to use such synthesized long documents as inputs. Moreover, this dataset is evidently more beneficial to models trained on these synthesized short texts, therefore introducing a potential bias.

2. The experimental results can not convince me.  The authors claim that when we scale up the input context from 1k tokens to 8k tokens, the results continuously decrease which is interesting. However, when the answer does not appear at the beginning of the document, gpt3.5-16k usually outperforms gpt3.5-4k.
I think the potential reasons are as follows:
(1) Unfair metrics. This paper mainly uses n-gram matching metrics like ROUGE and F-1 which may not correlate with human evaluation results.
(2) Position bias in the dataset construction. Most answers are at the beginning of the document, additional input is noise.
(3) The current long context models indeed cannot handle many input tokens but how is the Llama2-4k?

3. The authors simply left the main results in the appendix without analysis. For example, I find that in many tasks even a 6B model can outperform gpt3.5 by a remarkable margin. More analysis can make us better understand current long context models.

**Questions:**

see weaknesses

---

> ### Author Response · Authors · 2023-11-16
>
> Thank you for your valuable feedback and insightful observations regarding our paper.  We appreciate the opportunity to address your concerns and clarify aspects of our work. Please find our detailed responses below:
>
> **Q1: The construction of the dataset is detached from realistic scenarios.**
>
> We acknowledge your concern regarding the realistic applicability of our dataset.
> It is important to note that M4LE includes 10 tasks that directly target global context understanding abilities pertinent to real-world scenarios, such as summarization, translation, question answering, and classification tasks. These tasks represent a significant portion of our benchmark and are designed to reflect common use cases of LLMs in practical applications.
>
> Furthermore, while some tasks in M4LE may seem detached from typical user interactions with LLMs, they are intentionally designed to evaluate LLM across five key long-context understanding abilities as mentioned in the paper. These five abilities encompass a broad spectrum of long-sequence scenarios and challenges that a language model may face in real-world applications. Therefore, by constructing a diverse benchmark that evaluates these five abilities, we can obtain a measure that closely approximates the performance of models in realistic, long-context scenarios.
>
> **Q2: This dataset is evidently more beneficial to models trained on these synthesized short texts, therefore introducing a potential bias.**
>
> We understand the concern about potential biases favouring models trained on synthesized short texts. However, this aspect was one of the motivations behind our construction methodology. While the models might be trained on the original dataset, combining the shorter texts presents an instance that is sufficiently distinct from the original ones. For instance, in the WikiText-103 (NLI) task, we observed that even though Wikipedia articles are commonly included in language model training datasets, the performance of most models is poor even at shorter context lengths. This suggests that the benchmark effectively challenges the models beyond their training conditions.
>
> Moreover, the primary objective of M4LE is to assess the ability of LLMs to maintain their performance as context length increases. By focusing on performance trends across varying context lengths, we minimize the impact of any prior training on shorter texts.
>
> **Q3: However, when the answer does not appear at the beginning of the document, gpt3.5-16k usually outperforms gpt3.5-4k. I think the potential reasons are as follows: (1) Unfair metrics. This paper mainly uses n-gram matching metrics like ROUGE and F-1 which may not correlate with human evaluation results. (2) Position bias in the dataset construction. Most answers are at the beginning of the document, additional input is noise. (3) The current long context models indeed cannot handle many input tokens but how is the Llama2-4k?**
>
>
> Regarding the evaluation metrics, we use accuracy as the metric for QA, consistent with previous work (Liu et al., 2023). For classification tasks, we use F1 scores, which do not rely on n-gram matching. N-gram metrics are applied in only 15 of the 36 tasks, primarily in summarization and translation. We acknowledge the limitations of these metrics but have focused on performance trends over context lengths to mitigate potential biases. Besides, these N-gram metrics are still commonly reported in much recent literature for summarization and translation tasks.
>
> To address potential position biases, we have carefully constructed instances with relevant supporting paragraphs uniformly distributed within the input context, as outlined in Section 3.1 of our paper.
>
> Finally, regarding Llama2-4k, we used dynamic NTK scaling to extend the effective window size of LLama2 models, which we found to be an effective approach without further fine-tuning as mentioned in the paper.
>
> **Q4: I find that in many tasks even a 6B model can outperform gpt3.5 by a remarkable margin. More analysis can make us better understand current long context models.**
>
> Thank you for your observations and suggestions! GPT-3.5 outperforms other models in most tasks, except for DuReader and CNNNews where ChatGLM2-6b-32K performs better. We examined some of the outputs generated by different models in various tasks and did not find anything insightful. We believe that other models may occasionally outperform GPT-3.5 due to the inherent noise in extensive experiments.
> Overall, GPT-3.5 exhibits strong long-context understanding, but certain tasks might favour other models due to random noise factors.
>
> In summary, we believe that M4LE provides a robust and comprehensive benchmark for evaluating the long-context understanding abilities of LLMs, addressing both realistic scenarios and potential biases. We hope our response adequately addresses your concerns and we are open to further discussion or clarification if needed.

---

> ### Author Response · Authors · 2023-11-23
>
> Thank you for your valuable time and feedback. As the deadline for author-reviewer discussion ends today, we kindly request that you acknowledge our rebuttal and engage in further discussion if you have any additional comments.

---

### Author Response · Authors · 2023-11-16

We greatly appreciate the insightful feedback and constructive criticisms from all the reviewers provided for our submission. Your comments have been helpful in enhancing the clarity, precision, and overall quality of our work.

Thank you for all the valuable suggestions and comments.
We would like to emphasize that M4LE is the first comprehensive benchmark for evaluating the five crucial abilities relevant to long-context understanding, encompassing a wide array of practical scenarios, accommodating contexts extend up to 32K words (even more in future versions).
Given that recent LLMs can already support context length of 100K words or more, we believe our approach offers a significant contribution to create challenging long-context evaluation data that adequately represents these five key abilities. Though some tasks might appear somewhat abstracted from everyday scenarios, they are deliberately chosen to target distinct abilities. This approach ensures a multifaceted assessment of LLMs' proficiency in processing lengthy inputs as encountered in practical situations.

We would also like to announce that we have updated the results of experiments that encountered memory problems, particularly for contexts over 6K words. In the initial submission, as mentioned in the "Computational Resources Constraint" section of the Appendix, we employed input truncation from the left in cases of GPU memory limitations. We found this approach adversely affected many models' performance. To rectify this, we have re-conducted these experiments with enhanced GPU resources, ensuring a more accurate and fair evaluation of the models. All relevant figures, tables and analyses in the manuscript have been updated to reflect these changes.

---

### Meta-Review · Area_Chair_mqeM · 2023-12-13

**Metareview:**

The paper introduces a benchmark consisting of a wide range of tasks for evaluating the abilities of language models for long range tasks. The benchmark includes 5 different abilities and context length ranges.
The main findings is that several LLMs seem to struggle with long context understanding.

In general the reviewers appriciate the motivation of the paper. The other strengths commonly mentioned by the reviewers include the extensiveness of the collected data (ni6x, 2aid, seiz), and the extensiveness of the evaluations (ni6x, qzry).

However, the reviewers identified several key weaknesses. Most reviewers remain skeptical about the usefulness of the dataset given that it is synthetically constructed from short texts. The observations from evaluation does _not_ seem very insightful and some findings are found to be confusing (qZry, sezj). While some of the clarity issues are resolved during discussion period, other concerns remained unresolved.

**Justification For Why Not Higher Score:**

There seem to be several major concerns regarding the lack of insights and the overall execution, especially with the core dataset construction method, which involves synthetically creating long-range datasets from short text inputs.

**Justification For Why Not Lower Score:**

N/A

---

### Decision · Program_Chairs · 2024-01-16

Reject